# Three SARS-CoV-2 spike protein variants delivered intranasally by measles and mumps vaccines are broadly protective

Yuexiu Zhang[1,10], Michelle Chamblee[1,10], Jiayu Xu[1,10], Panke Qu[1], Mohamed M. Shamseldin[2,3,4], Sung J. Yoo[1], Jack Misny[5], Ilada Thongpan[5], Mahesh KC[5], Jesse M. Hall[2], Yash A. Gupta[2], John P. Evans[1], Mijia Lu[1], Chengjin Ye[6], Cheng Chih Hsu[1], Xueya Liang[1], Luis Martinez-Sobrido[6], Jacob S. Yount[2,7], Prosper N. Boyaka[1,7], Shan-Lu Liu[1,2,7,8], Purnima Dubey[2,7], Mark E. Peeples[5,7,9] & Jianrong Li[1,7] ✉

As the new SARS-CoV-2 Omicron variants and subvariants emerge, there is an urgency to develop intranasal, broadly protective vaccines. Here, we developed highly efficacious, intranasal trivalent SARS-CoV-2 vaccine candidates (TVC) based on three components of the MMR vaccine: measles virus (MeV), mumps virus (MuV) Jeryl Lynn (JL1) strain, and MuV JL2 strain. Specifically, MeV, MuV-JL1, and MuV-JL2 vaccine strains, each expressing prefusion spike (preS-6P) from a different variant of concern (VoC), were combined to generate TVCs. Intranasal immunization of IFNAR1[−/−] mice and female hamsters with TVCs generated high levels of S-specific serum IgG antibodies, broad neutralizing antibodies, and mucosal IgA antibodies as well as tissue-resident memory T cells in the lungs. The immunized female hamsters were protected from challenge with SARS-CoV-2 original WA1, B.1.617.2, and B.1.1.529 strains. The preexisting MeV and MuV immunity does not significantly interfere with the efficacy of TVC. Thus, the trivalent platform is a promising next-generation SARS-CoV-2 vaccine candidate.

Coronavirus disease 2019 (COVID-19) caused by severe acute respiratory syndrome coronavirus 2 (SARS-CoV-2) has led to over 775 million cases and over 7.0 million deaths worldwide as of May of 2024 according to the WHO. Since 2021, several SARS-CoV-2 vaccines including the Pfizer and Moderna mRNA vaccines, the Janssen Ad26-vectored vaccine, and the Novavax subunit vaccine have been authorized for intramuscular administration in humans. These vaccines effectively prevent severe disease, hospitalization, and death associated with SARS-CoV-2 but do not prevent infection and transmission as efficiently[1–3]. They all utilize the prefusion form of the spike (S) protein, stabilized by 2 proline mutations (preS-2P)[4,5], which induces more effective neutralizing antibodies (NAbs) than the native S protein[4,6].

As the pandemic continues, new SARS-CoV-2 variants and subvariants emerge, each containing mutations that enhance transmissibility, disease severity, antibody escape and/or immune evasion[7].

[1]Department of Veterinary Biosciences, The Ohio State University, Columbus, OH, USA. [2]Department of Microbial Infection and Immunity, College of Medicine, The Ohio State University, Columbus, OH, USA. [3]Department of Microbiology, The Ohio State University, Columbus, OH, USA. [4]Department of Microbiology and Immunology, Faculty of Pharmacy, Helwan University, Ain Helwan, Helwan, Egypt. [5]Center for Vaccines and Immunity, Abigail Wexner Research Institute at Nationwide Children's Hospital, Columbus, OH, USA. [6]Texas Biomedical Research Institute, San Antonio, TX, USA. [7]Infectious Disease Institute, The Ohio State University, Columbus, OH, USA. [8]Center for Retrovirus Research, The Ohio State University, Columbus, OH, USA. [9]Department of Pediatrics, College of Medicine, The Ohio State University, Columbus, OH, USA. [10]These authors contributed equally: Yuexiu Zhang, Michelle Chamblee, Jiayu Xu. ✉e-mail: li.926@osu.edu

Previously dominant circulating variants of concern (VoCs) such as Alpha (B.1.1.7), Beta (B.1.351), Delta (B.1.617.2), Omicron (BA.1, B.1.1.529), and Omicron subvariants (BA.2, BA.3, BA.4, BA.5, BQ.1.1, XBB.1.5, EG.5, and HV.1), have led to waves of new COVID-19 cases. Since January of 2024, a new subvariant, JN.1, has become the dominant virus[8]. The SARS-CoV-2 WA1 S-based vaccines are ineffective against these Omicron variant and subvariants[8,9]. This has led to the development of a bivalent mRNA vaccine booster composed of preS-2P of the original SARS-CoV-2 WA1 strain and Omicron subvariant BA.4/5[10,11]. However, the emergency of Omicron XBB.1.5 dramatically reduced the efficacy of the bivalent mRNA vaccine[8,12]. Recently, a monovalent mRNA vaccine expressing preS-2P of Omicron subvariant XBB.1.5 was approved to prevent infection against XBB.1.5 and its relatives[13,14]. Though there is an increase in protection with these boosters, there remains a lack of mucosal immunity generated by these vaccines[15–17]. Mucosal immunity, including IgA antibodies in the respiratory tract, provides a first line of protection against respiratory diseases such as SARS-CoV-2[16]. A major goal of the next generation of SARS-CoV-2 vaccines is intranasal delivery[18]. With the rapid evolution of new Omicron subvariants and the co-circulation of multiple Omicron subvariants, there is an urgent need for a rapidly adaptable vaccine that can provide broad protection against multiple VoCs and Omicron subvariants.

In the late 1960s, a live attenuated trivalent MMR (measles, mumps, and rubella) vaccine that is capable of providing long-term protection against the measles (MeV), mumps (MuV), and rubella viruses was developed[19]. It is provided as a two dose vaccine administered by injection at 9–15 months of age and again at 15 months to 6 years of age[19]. It has been one of the most successful vaccines with two doses being 97% effective against measles and 88% effective against mumps according to the US CDC, providing lifelong protection to vaccinees[20]. The MMR vaccine developed by Merck is composed of one MeV vaccine strain (Edmonston), two MuV vaccine strains [a major component Jeryl Lynn 1 (JL1) strain and a minor component JL2 strain][21,22]. MeV and MuV are both non-segmented negative-sense RNA viruses belonging to the family *Paramyxoviridae* and have since been utilized as effective viral vectors to deliver experimental vaccines or oncolytic gene therapy[23–25]. Although Merck and the US CDC recommend the subcutaneous immunization route for infants and children, early clinical trials suggested that intranasal immunization induced better NAb titers against MeV or MuV compared to the subcutaneous or intramuscular route[26–28]. A major advantage of intranasal immunization is that it can induce both systemic and mucosal immunity[27,29]. All currently approved SARS-CoV-2 vaccines are delivered intramuscularly, triggering strong peripheral serum NAbs but not mucosal antibodies in the respiratory tract[15,16]. Here, we have focused our efforts on developing next-generation intranasal SARS-CoV-2 vaccines to enhance protection of the respiratory tract, the initial/primary site of SARS-CoV-2 infection.

Since the pandemic began, our laboratory has been utilizing several non-segmented negative-sense RNA viruses including vesicular stomatitis virus (VSV)[30,31], MuV[32,33], and MeV[33–35] as vectors to deliver prefusion S protein stabilized by 6 prolines (preS-6P or HexaPro), which is more stable and has higher protein expression compared to preS-2P[36]. Specifically, we showed that preS-6P/HexaPro induces 2-4-fold more neutralizing antibodies against SARS-CoV-2 VoCs than the preS-2P when they were delivered by a VSV vector[30]. Similarly, in rMuV[32] or rMeV[35] vector, preS-6P induces 8.5 times higher NAbs compared to preS-2P. These studies highlight the importance of using preS-6P as the immunogen for the development of next generation SARS-CoV-2 vaccines.

Here, we have utilized three components of the MMR vaccine, MeV Edmonston, MuV JL-1, and MuV JL-2 as the vectors to deliver preS-6P proteins of the original SARS-CoV-2 WA1 and several VoCs (B.1.17, B.1.351, B.1.617.2, and B.1.1.529) and to develop intranasal trivalent SARS-CoV-2 vaccine candidates.

## Results

### Characterization of rMeV expressing preS-6P
We previously found that the prefusion S protein of SARS-CoV-2 stabilized by six prolines (preS-6P) rather than two prolines (preS-2P) is more immunogenic[30,32,35]. Thus, the *preS-6P* gene of SARS-CoV-2 WA1 or VoCs (B.1.351, B.1.1.7, and B.1.617.2) was inserted individually into the genome of the MeV Edmonston vaccine strain at the P-M gene junction using a yeast-based recombinant system (Fig. S1A). All recombinant viruses were recovered using the MeV reverse genetics system and were named rMeV-WA1, rMeV-B.1.351, rMeV-B.1.1.7, and rMeV-B.1.617.2. All four recombinant viruses formed smaller plaques compared to the parental rMeV (Fig. S1B).

Next, we examined the expression of preS-6P by MeV vector. A 180 kDa preS-6P protein was detected in cell lysates as well as in cell culture supernatants from rMeV-WA1-, rMeV-B.1.351-, rMeV-B.1.1.7-, or rMeV-B.1.617.2-infected Vero CCL81 cells, but not in the parental rMeV-infected cells (Fig. S1C). Thus, the soluble preS-6P proteins (lacking CT/TM) of SARS-CoV-2 WA1 and VoCs were highly expressed by the MeV and were secreted into the cell culture medium. All four rMeVs expressing preS-6P had a significant delay in syncytia formation compared to the parental rMeV (Fig. S2) but grew to similar titers in Vero CCL81 cells (Fig. S1D).

### Characterization of rMuV-JL2 expressing preS-6P
Using a similar strategy, the *preS-6P* gene was inserted into the genome of the MuV-JL2 strain at the P-M gene junction and recombinant rMuV-JL2 viruses expressing preS-6P proteins of SARS-CoV-2 B.1.351, B.1.1.7, and B.1.617.2 were recovered (Fig. S3A). All recombinant rMuV-JL2 with *preS-6P* insertions formed smaller plaques compared to the parental MuV-JL2 (Fig. S3B). The preS-6P protein was detected in both cell culture supernatant and lysate in rMuV-JL2-WA1, rMuV-JL2-B.1.351, rMuV-JL2-B.1.1.7, or rMuV-JL2-B.1.617.2-infected Vero CCL81 cells, but not the parental rMuV-JL2-infected cells (Fig. S3C). The parental rMuV-JL2 had massive syncytia formation at day 2 post-infection whereas all rMuV-JL2 expressing preS-6P showed maximal cytopathic effects (CPE) at day 4 (Fig. S4). All recombinant viruses grew to similar titers in Vero CCL81 cells (Fig. S3D).

### Characterization of rMuV-JL1 expressing preS-6P
Similar to MuV-JL2, we generated three recombinant rMuV-JL1 viruses expressing preS-6P proteins (rMuV-JL1-WA1, rMuV-JL1-B.1.1.7, and rMuV-JL1-B.1.617.2) (Fig. S5A). All three rMuV-JL1 expressing preS-6P formed significantly smaller plaques compared to the parental rMuV-JL1 (Fig. S5B). A similar level of preS-6P protein was detected in rMuV-JL1-WA1, rMuV-JL1-B.1.1.7, or rMuV-JL1-B.1.617.2-infected Vero CCL81 cells, but not rMuV-JL1-infected cells (Fig. S5C). Recombinant rMuV-JL1-WA1, rMuV-JL1-B.1.1.7, and rMuV-JL1-B.1.617.2 exhibited delayed CPE compared to the parental MuV-JL1 (Fig. S6). All recombinant viruses grew to similar titers in Vero CCL81 cells (Fig. S5D).

### Strategy for formulation of trivalent vaccine candidates (TVC)
We next combined an equal amount (PFU) of MeV, MuV-JL1, and MuV-JL2 vaccine strains, each expressing the preS-6P of original SARS-CoV-2 WA1 or VoCs, to formulate trivalent vaccine candidates (TVC) (Table S1). For vector controls, we combined equal amounts of rMeV, rMuV-JL1, and rMuV-JL2 (MMM vector). The rationale for testing the selected TVCs in mouse and hamster models is summarized in Table S2.

### Trivalent vaccine candidates are highly immunogenic in a mouse model
We first tested the immunogenicity of TVC-I (rMuV-JL2-WA1, rMuV JL2-B.1.1.7, and rMeV-B.1.351), TVC-II (rMuV-JL2-WA1, rMuV-JL2-B.1.1.7, and rMeV-WA1), and a monovalent rMuV-JL2-WA1 in IFNAR1$^{-/-}$ mice, which are susceptible to MeV and MuV infection (Fig. 1A)[32,37,38]. At week 7, serum IgG titers were determined by ELISA using the preS-6P proteins

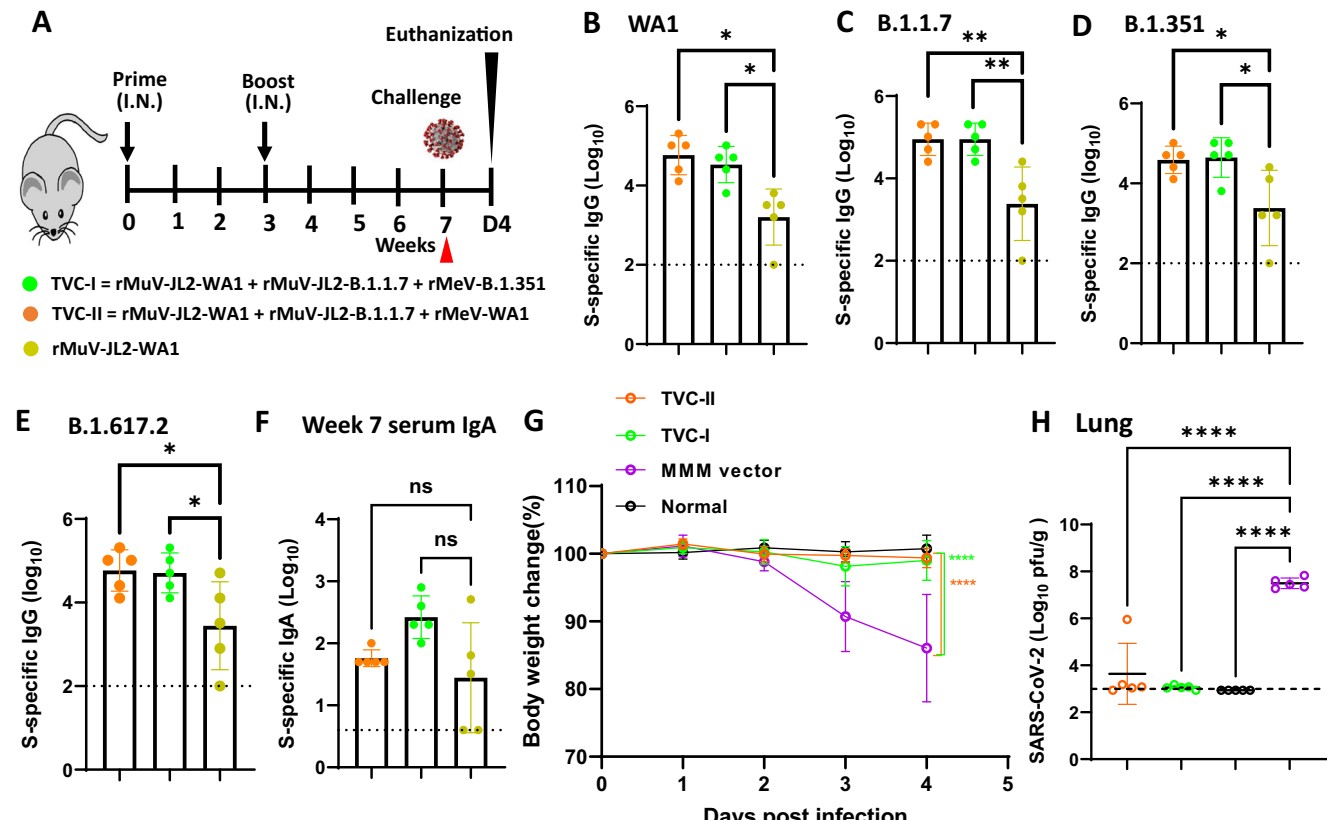

**Fig. 1 | Immunogenicity of trivalent vaccine candidates (TVC-I and TVC-II) in IFNAR1$^{-/-}$ mice. A** Schematic diagram of mice immunization, sample collection, and challenge. IFNAR1$^{-/-}$ mice ($n = 5$ per group) were immunized intranasally with $1.2 \times 10^6$ PFU of TVC-I (a mixture of rMuV-JL2-WA1, rMuV JL2-B.1.1.7, and rMeV-B.1.351, $4 \times 10^5$ PFU per virus), TVC-II (a mixture of rMuV-JL2-WA1, rMuV-JL2-B.1.1.7, and rMeV-WA1, $4 \times 10^5$ PFU per virus), rMuV-JL2-WA1, or MMM vector control, and were boosted 3 weeks later. At week 7, sera were collected for detection of S-specific IgG titer using the preS-6P protein of SARS-CoV-2 WA1 (**B**), B.1.1.7 (**C**), B.1.351 (**D**), or B.1.617.2 (**E**) as ELISA coating antigen. (**F**) Serum IgA titer at week 7. The ELISA was conducted using the preS-6P protein of SARS-CoV-2 WA1 as the coating antigen. IgG and IgA titers are the geometric mean titer (GMT) of 5 mice ± standard deviation (SD). The *P*-value of TVC-I and TVC-II vs rMuV-JL2-WA1 in (**B**) is *$P = 0.0327$ and *$P = 0.0116$, **C** is *$P = 0.0017$ and *$P = 0.0017$, **D** is *$P = 0.0219$ and *$P = 0.0292$, and **E** is *$P = 0.0472$ and *$P = 0.0361$. **G** Changes in body weight of mice. At week 7, mice were challenged with SARS-CoV-2 MA10, and body weight was measured daily until day 4. Normal refers to animals inoculated with DMEM. Percent of initial body weight is shown. Data are the average of 5 mice ($n = 5$) ± SD. The *P*-value of TVC-I and TVC-II vs MMM vector is ****$P = 2.893 \times 10^{-7}$ and ****$P = 2.03 \times 10^{-8}$, respectively. **H** Viral burden in the lung. At day 4 post-challenge, SARS-CoV-2 MA10 titer in the lungs was measured by plaque assay. Data shown are the GMT of 5 mice ($n = 5$) ± SD. The *P*-value of TVC-I, TVC-II, and normal control vs MMM vector is ****$P = 6.36 \times 10^{-8}$, ****$P = 4.414 \times 10^{-7}$, and ****$P = 4.35 \times 10^{-8}$, respectively. The dotted line indicates the limit of detection (LoD) which is 2.7 Log$_{10}$ PFU per gram of tissue. Statistical analyses in (**B**–**F**) and (**H**) were conducted using one-way ANOVA. Statistical analyses in (**G**) were conducted using two-way ANOVA. (*$P < 0.05$; **$P < 0.01$; ***$P < 0.001$; ****$P < 0.0001$; ns, not significant). Source data are provided in the Source Data file.

of SARS-CoV-2 WA1 (Fig. 1B), B.1.1.7 (Fig. 1C), B.1.351 (Fig. 1D), and B.1.617.2 (Fig. 1E) as a coating antigen. Both TVC-I and TVC-II induced significantly higher serum IgG titers than the monovalent rMuV-JL2-WA1 (Fig. 1B–E). However, TVC-I and TVC-II induced similar levels of serum IgA compared to rMuV-JL2-WA1 at week 7 (Fig. 1F).

At week 7, mice in the TVC-I, TVC-II, and MMM vector groups were challenged with a mouse-adapted (MA) SARS-CoV-2 (strain MA10). Mice in the MMM vector control lost ~15% of weight by day 4, and succumbed (Fig. 1G). In contrast, mice immunized with either trivalent vaccine had no weight loss. The lungs from the MMM vector group had an average titer of 7.2 log$_{10}$ PFU/g tissue (Fig. 1H). MA SARS-CoV-2 titers in the trivalent vaccine groups were near or below the detection limit (Fig. 1H). Thus, TVC-I and TVC-II were highly immunogenic and provided complete protection against challenge with SARS-CoV-2 MA10.

### Intranasal immunization is a superior immunization route

We chose TVC-III (rMuV-JL1-WA1 + rMuV-JL2-B.1.617.2 + rMeV-B.1.351) (Table S2) to compare the efficacy of three different immunization routes: intranasal (I.N.), subcutaneous (S.C.), and a combination (I.N.+S.C.) of intranasal and subcutaneous. In all cases, sera from weeks 5 and 7 in the I.N. group induced significantly higher serum IgG compared to the S.C. or the I.N.+S.C. groups (Fig. 2A–C). However, there was no significant difference in serum IgG between the S.C. and the I.N.+S.C. groups (Fig. 2A–C).

Sera at week 7 were used for the determination of SARS-CoV-2-specific NAb using a lentivirus-pseudotyped neutralization assay[39]. Sera from the I.N. group had average NAb titers of 3,013, 1,821, and 2,138, and 309 against pseudotyped lentivirus bearing the S protein with the WA1 (D614G), B.1.1.7, B.1.351, and B.1.617.2, respectively (Fig. 2D). However, NAb titers against Omicron subvariants (BA.1 and BA.4/5) were barely detectable (Fig. 2D). A similar pattern was observed in the S.C. and the I.N.+S.C. groups. Among these three immunization routes, I.N. induced the highest NAbs, I.N.+S.C. was the second best, and S.C. induced the lowest NAbs (Fig. 2D).

Mice in the I.N. group produced significantly higher IgA than those in the I.N.+S.C. group (Fig. 2E–G). As expected, lung IgA titer in the S.C. group was below the detection limit (Fig. 2E–G). In addition, IgG titers in the I.N. and I.N.+S.C. groups were higher than those in the S.C. group but the difference was not significant (Fig. 2H–J). Thus, I.N. induces the highest mucosal IgA response and I.N.+S.C. induces a

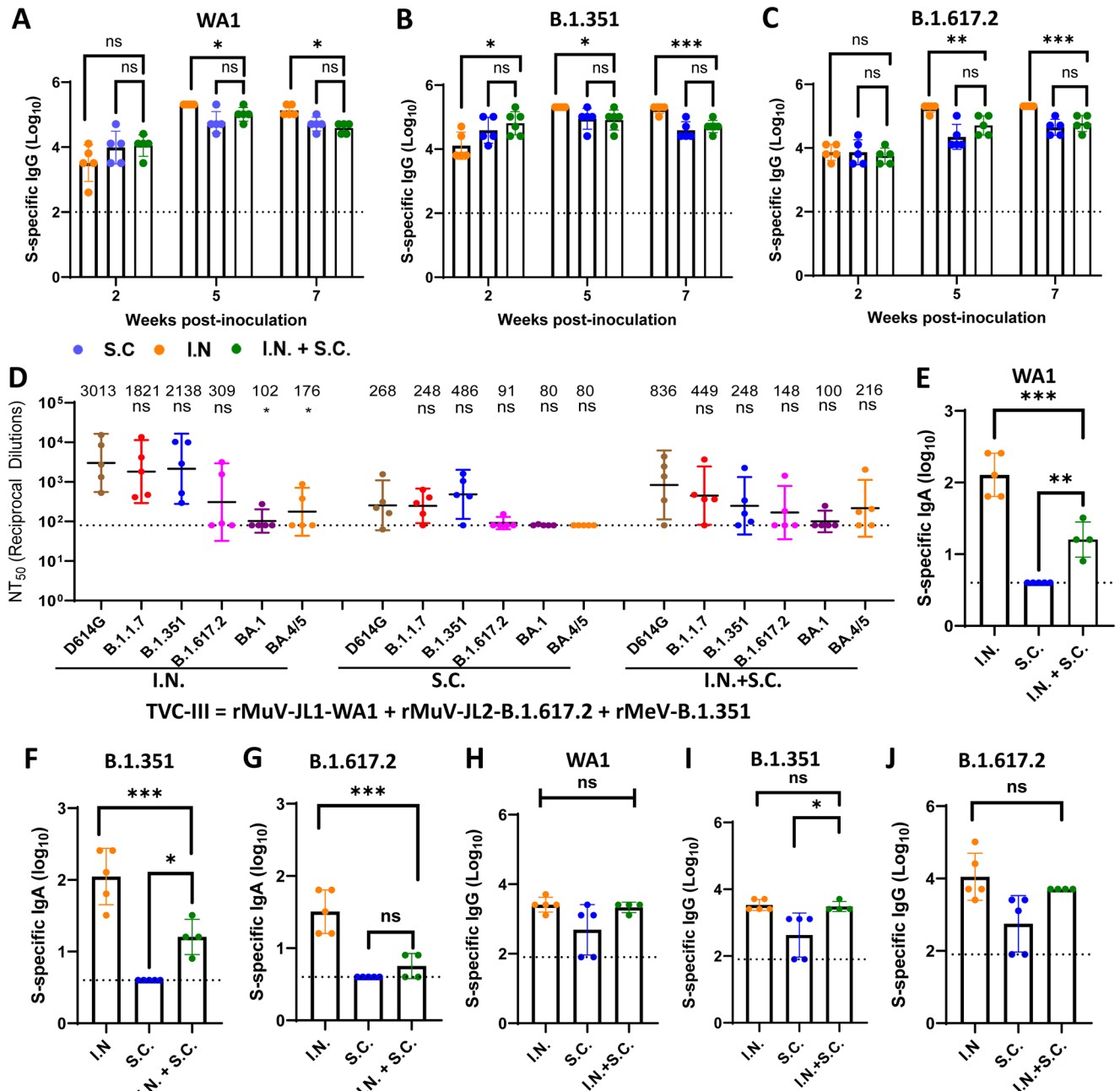

**Fig. 2 | Intranasal immunization of trivalent vaccine (TVC-III) is superior to subcutaneous or a combination of intranasal and subcutaneous route.** IFNAR1$^{-/-}$ mice were immunized with $1.2\times10^6$ PFU of TVC-III (rMuV-JL1-WA1 + rMuV-JL2-B.1.617.2 + rMeV-B.1.351) via I.N., S.C., or I.N. + S.C., and were boosted via the same route three weeks later. At weeks 2, 5, and 7, serum was collected for detection of S-specific IgG titer by ELISA using preS-6P protein of SARS-CoV-2 WA1 (**A**), B.1.351 (**B**), or B.1.617.2 (**C**) as the coating antigen. The P-value for I.N. vs I.N. + S.C. in (**A–C**) is: **A**, \*P = 0.013349 (week 5) and \*P = 0.013349 (week 7); **B**, \*P = 0.013349 (week 2), \*P = 0.039969 (week 5), and \*\*\*P = 0.000478 (week 7); **C**, \*\*P = 0.006271 (week 5) and \*\*\*P = 0.000478 (week 7). Sera at week 7 were used for detection of SARS-CoV-2 NAbs using a lentivirus pseudotyped neutralization assay against SARS-CoV-2 WA1 (D614G), B.1.1.7, B.1.351, B.1.617.2, Omicron BA.1 (B.1.1.529), or BA.4/5 spike. The 50% neutralization titer (NT$_{50}$) was calculated for each serum sample (**D**). Data are the mean of five mice (n = 5) ± SD. The P-value for BA.1 and BA.4/5 vs WA1 (D614G) is \*P = 0.0426 and \*P = 0.0279, respectively. At week 7, mice were euthanized, and BAL was collected from the lungs of each mouse for detection of IgA titer by ELISA using the preS-6P protein of SARS-CoV-2 WA1 (**E**), B.1.351 (**F**), or B.1.617.2 (**G**), and for detection of IgG titer by ELISA using the preS-6P protein of SARS-CoV-2 WA1 (**H**), B.1.351 (**I**), or B.1.617.2 (**J**). All antibody titers are the GMT of 5 or 4 mice (n = 5 or 4) ± SD. The dotted line indicates the limit of detection. In **E**, the P-value for I.N. and S.C. vs I.N. + S.C. is \*\*\*P = 0.0002 and \*\*P = 0.0036, respectively. In **F**, the P-value for I.N. and S.C. vs I.N. + S.C. is \*\*\*P = 0.0013 and \*P = 0.0121, respectively. In **G**, the P-value for I.N. vs I.N. + S.C. is \*\*\*P = 0.0003. In **I**, the P-value for S.C. vs I.N. + S.C. is \*\*\*P = 0.0194. Statistical analyses were conducted using one-way ANOVA (\*P < 0.05; \*\*P < 0.01; \*\*\*P < 0.001; \*\*\*\*P < 0.0001; ns not significant). Source data are provided in the Source Data file.

moderate IgA response whereas S.C. does not induce any IgA antibodies.

After euthanization, spleens from the S.C. and the I.N. groups were isolated and splenocytes were stimulated with SARS-CoV-2 S peptides for the detection of T cell immune responses. T cell subsets such as T helper cells (Th) function as activators of cytotoxic T cells and B cells to aid in combating pathogens by eliciting cytokines. The Th1 subset of helper T cells is responsible for targeting intracellular pathogens and eliciting signature cytokines such as IFN-γ and TNF-α. Both the I.N. and the S.C. groups were able to elicit significantly higher

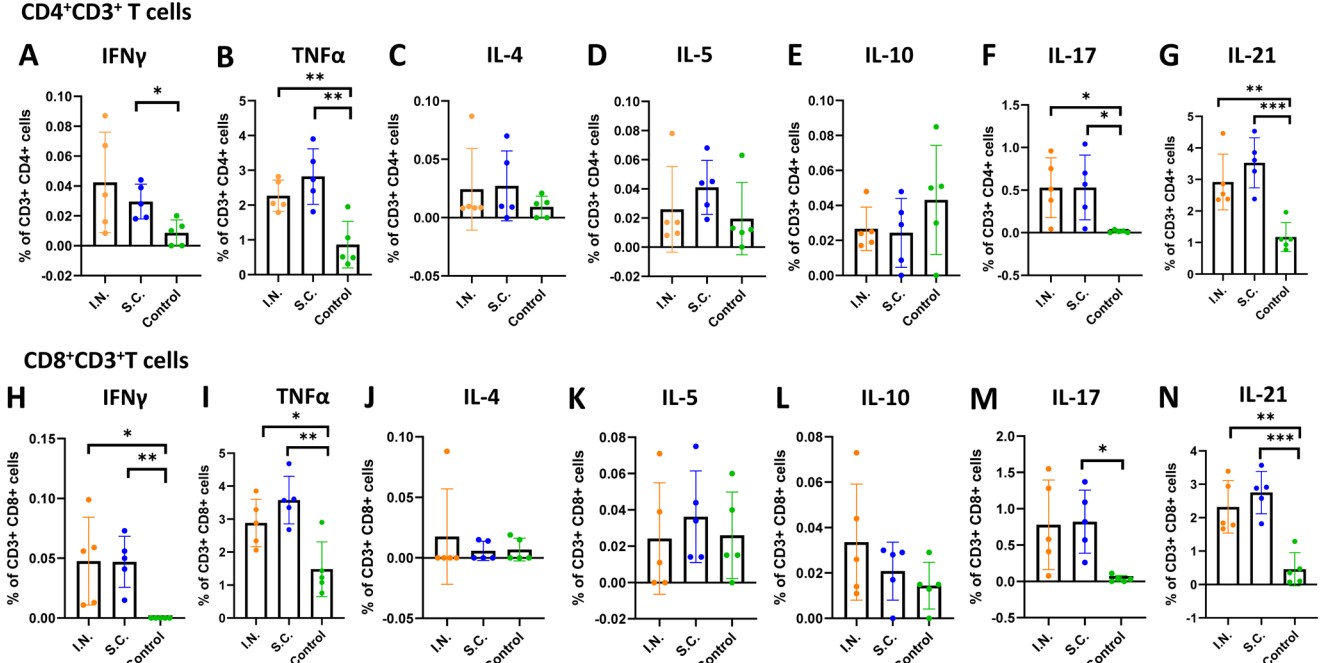

**Fig. 3 | Intranasal or subcutaneous immunization of trivalent vaccine induces a systemic T cell response.** Spleens from the intranasal and subcutaneous group in Fig. 2A–C were isolated and homogenized, and spleen T cell suspensions were prepared and seeded in three wells (triplicate per mouse) in 96-well plates and stimulated with peptide pool of SARS-CoV-2 WA1 S for 5 h. The frequencies of S-specific IFN-γ+CD4+ (**A**), TNF-α+CD4+ (**B**), IL-4+CD4+ (**C**), IL-5+CD4+ (**D**), IL-10+CD4+ (**E**), IL-17+ CD4+ (**F**), IL-21+ CD4+ (**G**), IFN-γ+CD8+ (**H**), TNF-α+CD8+ (**I**), IL-4+CD8+ (**J**), IL-5+CD8+ (**K**), IL-10+CD8+ (**L**), IL-17+ CD8+ (**M**), and IL-21+CD8+ (**N**) cells were determined by flow cytometry after intracellular staining with the corresponding anti-cytokine antibody. Data are the mean of five mice ($n = 5$) ± SD. In **A**, the P-value for S.C. vs control is *$P = 0.0475$. In **B**, the P-value for I.N. and S.C. vs control is **$P = 0.0098$ and **$P = 0.003$, respectively. In **F**, the P-value for I.N. and S.C. vs control is *$P = 0.0339$ and *$P = 0.033$, respectively. In **G**, the P-value for I.N. and S.C. vs control is **$P = 0.0051$ and ***$P = 0.0005$, respectively. In **H**, the P-value for I.N. and S.C. vs control is *$P = 0.0178$ and **$P = 0.0012$, respectively. In **I**, the P-value for I.N. and S.C. vs control is *$P = 0.0232$ and **$P = 0.0017$, respectively. In **M**, the P-value for S.C. vs control is *$P = 0.0264$. In **N**, the P-value for I.N. and S.C. vs control is **$P = 0.0013$ and ***$P = 0.0002$, respectively. Statistical analysis was conducted using one-way ANOVA (ns > 0.05, *$P < 0.05$; **$P < 0.01$; ***$P < 0.001$). Source data are provided in the Source Data file.

Th1 cytokines (IFN-γ and TNF-α) than the control ($P < 0.05$, or 0.01) for both CD4+ (Fig. 3A, B) and CD8+ (Fig. 3H, I) T cells. Th2 cells are characterized by eliciting cytokines such as IL-4, IL-5, and IL-10. There were no significant differences in Th2 cytokines elicited in the I.N., S.C., and control groups for CD4+ (Fig. 3C–E) or CD8+ (Fig. 3J–L) T cells. Follicular T helper cells (T$_{FH}$) that express IL-21 and T helper cells that express IL-17 (T$_H$17) play a role in B cell activation and differentiation. Cells from mice immunized I.N. and S.C. groups produced both of these cytokines. Mice immunized I.N. or S.C. expressed significantly higher IL-17 cytokines than the control group ($P < 0.05$) for both CD4+ (Fig. 3F) and CD8+ (Fig. 3M) T cells. However, there was no significant difference between the I.N. and S.C. group (Fig. 3F and M). In addition, mice immunized S.C. ($P = 0.0005$, $P = 0.0002$) or I.N. ($P = 0.0051$, $P = 0.0013$) elicited significantly higher IL-21 compared to the control group in both CD4+ (Fig. 3G) and CD8+ (Fig. 3N) T cells. These results show that both I.N. and S.C. groups are capable of eliciting CD4+ and CD8+ T cells towards a predominantly Th1/17 response.

**Intranasal immunization with TVC induces robust lung-resident T cells**

For many respiratory viruses, tissue-resident memory T cells (Trm) in lungs are critical for protection against repeated infection with the same virus or heterotypic variants that evade antibody responses[40,41]. We determined the induction of Trm in IFNAR1$^{-/-}$ mice intranasally immunized with a high ($1.2 \times 10^6$ PFU) or low ($3 \times 10^5$ PFU) dose of TVC-IV (rMeV-B.1.617.2, rMuV-JL1-B.1.1.7, and rMuV-JL2-WA1) or a high ($1.2 \times 10^6$ PFU) dose of monovalent rMuV-JL2-WA1 (Fig. 4A). Trivalent vaccine groups had higher WA1- (Fig. 4B), B.1.1.7- (Fig. 4C), B.1.351- (Fig. 4D), and B.1.617.2- (Fig. 4E) specific serum IgG than monovalent

vaccine but the difference was not significant. Similar result was observed for serum IgA titer (Fig. 4F).

At week 7, anti-CD45-PE was retro-orbitally injected into mice 10 min prior to euthanasia to separate the tissue-resident and circulating T cells in the lungs. After euthanasia, lung T cell suspensions were prepared and stimulated with SARS-CoV-2 S-specific peptide pool or myristate acetate (PMA)/ionomycin, and total lung CD4+ and CD8+ tissue-resident T cells (CD45$^-$) were analyzed.

Within the CD45$^-$ T cell population, the percentage of S-specific CD4+CD44+CD62L$^-$CD69$^+$ T cells (Fig. 5A and E) increased significantly in mice immunized with monovalent or trivalent vaccine compared to the MMM vector control. In addition, both low ($P = 0.0005$) and high ($P = 0.0074$) doses of trivalent vaccine groups had significantly higher percentage of CD45$^-$ T cells than the monovalent vaccine group. The IFN-γ–producing T cells (Fig. 5B and F) in the monovalent and trivalent vaccine groups were higher than the MMM vector control but the difference was not significant. The percentage of live S-specific CD4+CD69+ T cells in the high dose trivalent vaccine group was significantly higher than the MMM control ($P = 0.015$). The percentage of IL-17–producing cells significantly increased (Fig. 5C and G) in mice immunized with the trivalent vaccines either low or high dose compared to either the monovalent vaccine or the vector control. In addition, the trivalent vaccine groups induced higher IL-17–producing cells than the monovalent vaccine (Fig. 5C and G). The percent of S-specific IL-5-producing cells (Fig. 5D and H) was higher in monovalent and trivalent vaccine groups compared to the MMM control, but they were not statistically different. In addition, a low dose ($3 \times 10^5$ PFU) of trivalent vaccine included similar levels of T cell immune responses compared to the high dose ($1.2 \times 10^6$ PFU), suggesting that a dose of $3 \times 10^5$ PFU is sufficient to induce a higher level of Trm.

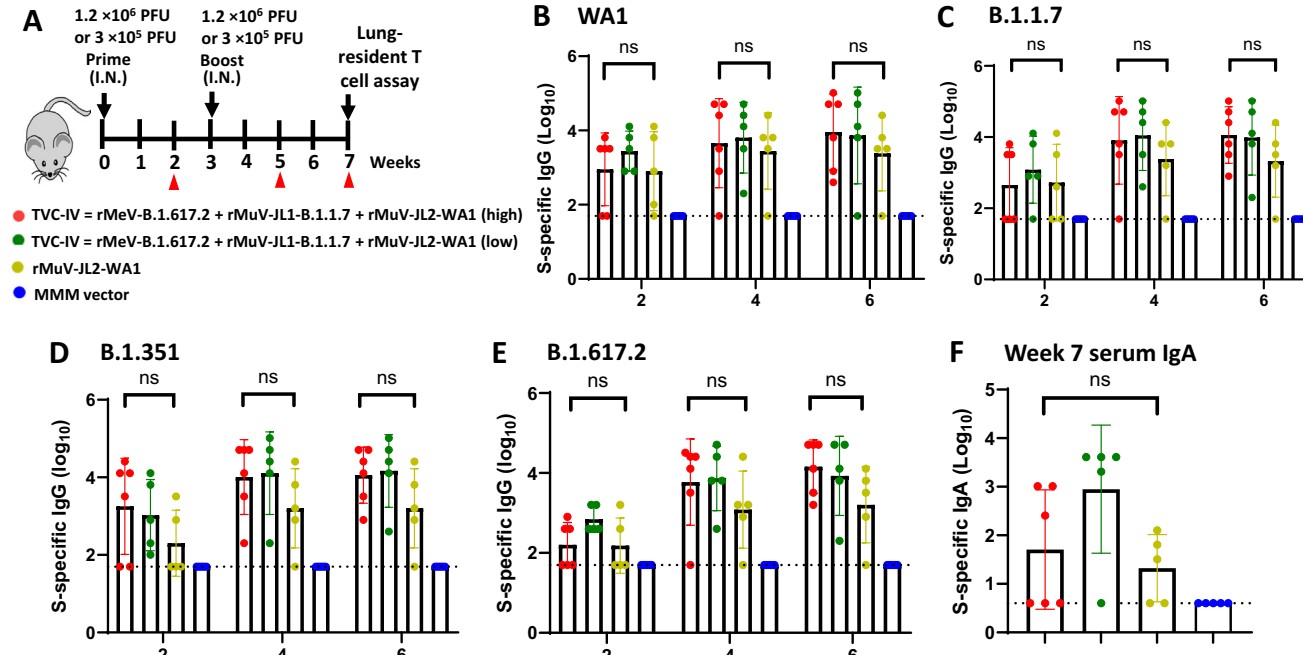

**Fig. 4 | Intranasal immunization of a low and a high dose of a trivalent vaccine (TVC-IV) induces a similar level of serum IgG and IgA. A** Immunization schedule in IFNAR1$^{-/-}$ mice. IFNAR1$^{-/-}$ mice ($n = 5$ or 6) were immunized intranasally with a high ($1.2 \times 10^6$ PFU) or a low ($3 \times 10^5$ PFU) dose of TVC-IV (rMeV-B.1.617.2, rMuV-JL1-B.1.1.7, and rMuV-JL2-WA1), a high dose ($1.2 \times 10^6$ PFU) of rMuV-JL2-WA1, or MMM vector, and boosted with the same dose 3 weeks later. At weeks 2, 5, and 7, serum was collected from each mouse and S-specific IgG titer was determined by ELISA using preS-6P protein of SARS-CoV-2 WA1 (**B**), B.1.1.7 (**C**), B.1.351 (**D**), or B.1.617.2 (**E**) as the coating antigen. In addition, sera at week 7 were used for the detection of serum IgA using preS-6P of SARS-CoV-2 WA1 as the coating antigen (**F**). IgG and IgA titers are the GMT of 5 or 6 mice ($n = 5$ or 6) $\pm$ SD. One-way ANOVA was used for statistical analyses. "ns" denotes no significant difference. Source data are provided in the Source Data file.

When CD45$^-$ CD4$^+$ T cells were stimulated with PMA/ionomycin, a similar but slightly different pattern was observed (Fig. S7). Specifically, the percentage of total CD4$^+$CD44$^+$CD62L$^-$CD69$^+$ cells (Fig. S7A and E) in trivalent vaccine groups were significantly higher than the vector or the monovalent vaccine group. IFN-γ producing Trms were increased in the trivalent and monovalent vaccine groups compared to the vector control (Fig. S7B and F), indicating that a Th1 polarization response was triggered by intranasal immunization. In addition, IL-17 (Fig. S7C and G), and IL-5 (Fig. S7D and H) producing Trms in trivalent vaccine groups were significantly higher than the vector control and monovalent vaccine groups. Therefore, trivalent vaccine groups induce more lung-resident CD4$^+$ T cells (predominantly Th1/17 polarized) than the monovalent vaccine group.

For tissue resident CD45$^-$CD8$^+$ T cells, the percentage of S-specific CD8$^+$CD44$^+$CD62L$^-$CD69$^+$ antigen–positive T cells (Fig. 5I, J) and IFN-γ producing T cells (Fig. 5K, L) were significantly higher in mice immunized with monovalent or trivalent vaccine compared to the MMM vector control. The trivalent vaccine induced more tissue resident CD45$^-$CD8$^+$ T cells than the monovalent vaccine but the difference was not statistically significant. A similar pattern (Fig. S7I–L) was observed when tissue resident CD45$^-$CD8$^+$ T cells were stimulated with PMA/ionomycin. Therefore, intranasally delivered trivalent and monovalent vaccines are capable of inducing antigen-specific lung-resident CD8$^+$ T cells.

### Trivalent vaccines protect hamsters against challenge with SARS-CoV-2 WA1 or VoCs
We next determined the immunogenicity of TVC-V (rMuV-JL1-WA1, rMuV-JL2-B.1.1.7, and rMeV-WA1), TVC-VI (rMuV-JL1-WA1, rMuV-JL2-B.1.617.2, and rMeV-WA1) (Table S2), and monovalent rMuV-JL2-B.1.617.2 in hamsters (Fig. 6A). Both TVC-V and TVC-VI induced

significantly higher serum IgG titers than the monovalent rMuV-JL2-B.1.617.2 using preS-6P of WA1 (Fig. 6B), B.1.1.7 (Fig. 6C), B.1.351 (Fig. 6D), or B.1.617.2 (Fig. 6E) as the ELISA coating antigens. Both TVC-V and TVC-VI induced significantly higher serum IgA titers than the monovalent rMuV-JL2-B.1.617.2 at weeks 5 (Fig. 6F) and 7 (Fig. 6G). Thus, TVC-V and TVC-VI are more immunogenic than the monovalent vaccine candidate.

Sera samples from TVC-V and TVC-VI groups were chosen to determine NAb titers. Both TVC-V and TVC-VI induced high NAb titers against WA1 (D614G) and B.1.1.7, moderate NAb titers against B.1.351 and B.1.617.2, and low NAb titer against Omicron BA.1 and BA.4/5 subvariants (Fig. 6H).

At week 7 post-immunization, 15 hamsters in the each of TVC-V and TVC-VI groups and MMM vector group were randomly divided into 3 subgroups ($n = 5$), which were challenged with original SARS-CoV-2 WA1, B.1.617.2, or B.1.1.529.

Following challenge with $2 \times 10^4$ PFU of SARS-CoV-2 WA1, the MMM vector control had ~5% weight loss, whereas the two trivalent vaccine groups had no weight loss (Fig. 7A). At day 4 post-challenge, the average SARS-CoV-2 titer in the lungs was 6.90 log$_{10}$ PFU/g tissue in the MMM vector group (Fig. 7B). In the TVC-V group, the average viral titer was 3.04 log$_{10}$ PFU/g tissue, which was near the detection limit (Fig. 7B). The viral titer in the lungs of the TVC-VI group was below the detection limit (Fig. 7B). In the nasal turbinate, the average viral titer in the MMM vector group was 7.27 log$_{10}$ PFU/g tissue, while the viral titer in both the TVC-V and TVC-VI groups was below the detection limit (Fig. 7C). The lung section from the MMM vector control had severe lung pathological lesions (average score of 2.9, Fig. 7D) such as interstitial pneumonia, pulmonary infiltration, edema, consolidation, and inflammation (Fig. S8). In contrast, lung pathology from both TVC-V and TVC-VI groups was mild (average score of 1.0, Fig. 7D) with minimal to moderate pulmonary

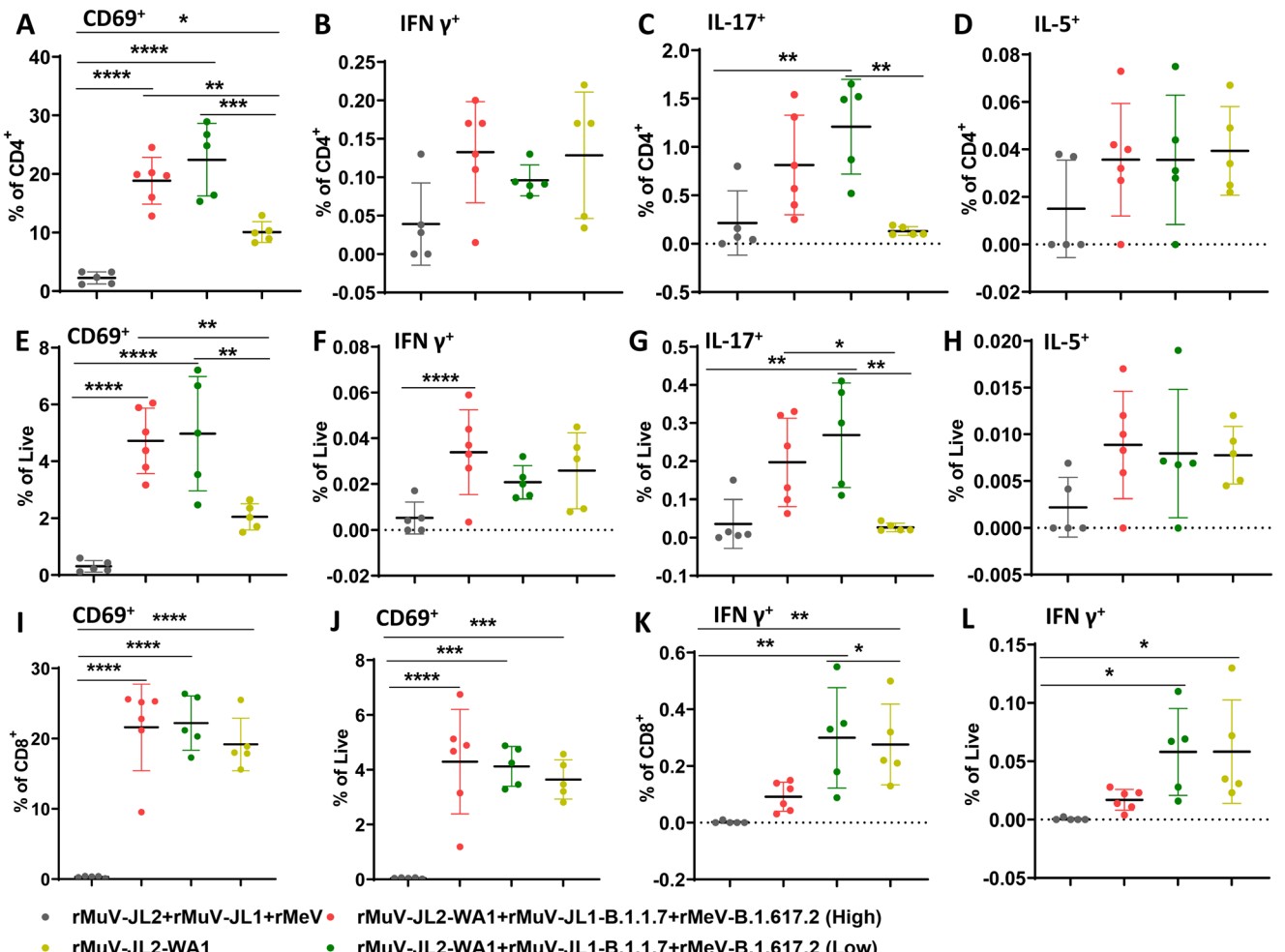

**Fig. 5 | Intranasal immunization of a trivalent vaccine induces superior tissue-resident memory T cells in the lungs to the monovalent vaccine.** At week 7, mice from Fig. 4 were retroorbitally injected with anti-CD45-PE. Ten minutes later, mice were euthanized, lung T cell suspensions were prepared and stimulated with SARS-CoV-2 S-specific peptide pool, and total lung $CD4^+$ and $CD8^+$ tissue-resident T cells ($CD45^-$) were analyzed. **A–D:** Percentage of $CD4^+$ of S specific $CD4^+CD69^+$ (**A**), IFN-γ (**B**), IL-17 (**C**), and IL-5 (**D**) $CD4^+$ T cells. **E–H:** Percentage of live cells of S specific $CD4^+CD69^+$ (**E**) IFN-γ (**F**), IL-17 (**G**), and IL-5 (**H**) $CD4^+$ T cells in the lung. **I-J:** Percentage of $CD8^+$ (**I**) and live (**J**) cells of S-specific $CD8^+CD69^+$ cells in the lung. **K-L:** Percentage of $CD8^+$ (**K**) and live (**L**) cells of S specific IFN-γ $CD8^+$ T cells in the lung. Data shown are the mean of 5 or 6 mice in each group ($n = 5$ or 6) ± SD. In **A**, the $P$-value for TVC-IV (High), TVC-IV (Low), and rMuV-JL2-WA1 vs MMM vector is ****$P = 9.01 \times 10^{-6}$, ****$P = 1.20 \times 10^{-6}$, and *$P = 0.0233$, respectively; TVC-IV (High) vs rMuV-JL2-WA1 is **$P = 0.0074$; and TVC-IV (Low) vs rMuV-JL2-WA1 is ***$P = 0.0005$. In **C**, the $P$-value for MMM vector vs TVC-IV (Low) is **$P = 0.0054$, and TVC-IV (Low) vs rMuV-JL2-WA1 is **$P = 0.0027$. In **E**, the $P$-value for TVC-IV (High) and TVC-IV (Low) vs MMM vector is ****$P = 5.84 \times 10^{-5}$ and ****$P = 5.01 \times 10^{-5}$, respectively; TVC-IV (High) and TVC-IV (Low) vs rMuV-JL2-WA1 is **$P = 0.0084$ and **$P = 0.0057$, respectively. In **F**, the $P$-value for TVC-IV (High) vs MMM vector is *$P = 0.015$. In **G**, the $P$-value for TVC-IV (Low) vs MMM vector is **$P = 0.0071$, TVC-IV (High) and TVC-IV (Low) vs rMuV-JL2-WA1 is *$P = 0.0438$ and **$P = 0.0052$, respectively. In **I**, the $P$-value for TVC-IV (High), TVC-IV (Low), and rMuV-JL2-WA1 vs MMM vector is ****$P = 1.247 \times 10^{-6}$, ****$P = 1.531 \times 10^{-6}$, and ****$P = 1.107 \times 10^{-5}$, respectively. In **J**, the $P$-value for TVC-IV (High), TVC-IV (Low), and rMuV-JL2-WA1 vs MMM vector is ****$P = 6.07 \times 10^{-5}$, ***$P = 0.0002$, and ***$P = 0.0006$, respectively. In **K**, the $P$-value for TVC-IV (Low) and rMuV-JL2-WA1 vs MMM vector is **$P = 0.0035$ and **$P = 0.0069$, respectively; TVC-IV (High) vs TVC-IV (Low) is *$P = 0.0352$. In **L**, the $P$-value for TVC-IV (Low) and rMuV-JL2-WA1 vs MMM vector is *$P = 0.0251$ and **$P = 0.0245$, respectively. One-way ANOVA with multiple comparisons was used to detect differences among groups (*$P < 0.05$; **$P < 0.01$; ***$P < 0.001$; ****$P < 0.0001$).

infiltration and inflammation (Fig. S8). Therefore, both trivalent vaccines provided complete protection against challenge with SARS-CoV-2 WA1.

After challenging with $2 \times 10^4$ PFU of B.1.617.2, hamsters in the MMM vector group had ~7% weight loss by day 4. In contrast, no weight loss was observed in either TVC-V or TVC-VI group (Fig. 7E). The MMM vector group showed an average titer of 6.0 and 6.2 $\log_{10}$PFU/g in the lung and nasal turbinate, respectively. In contrast, SARS-CoV-2 was undetectable in the lung and nasal turbinate in the TVC-V and TVC-VI groups (Fig. 7F, G). Severe lung pathology (average score of 3.3) was observed in all five lungs from the MMM vector control (Fig. 7H and Fig. S9). The TVC-V and TVC-VI groups had mild to moderate lung pathology (Fig. 7H and Fig. S9). Both trivalent vaccines provide complete protection against challenge with SARS-CoV-2 B.1.617.2 VoC.

For the Omicron BA.1 challenge, hamsters were intranasally administered $10^8$ PFU of Ad5-hACE2 and challenged 5 days later with $7 \times 10^5$ PFU SARS-CoV-2 B.1.1.529. None of the hamsters had significant weight loss ($P > 0.05$) (Fig. 7I). At day 3 post-challenge, the MMM vector control showed high viral loads with an average titer of 5.80 $\log_{10}$ PFU/g tissue (Fig. 7J). In contrast, the average viral titers in the TVC-V and TVC-VI immunized groups was 3.5 and 3.6 $\log_{10}$ PFU/g tissue, respectively (Fig. 7J). In the nasal turbinate, the MMM vector group showed an average titer of 6.37 $\log_{10}$ PFU/g tissue (Fig. 7K). However, the average titers in the TVC-V and TVC-VI groups were significantly lower (4.93 and 4.87 $\log_{10}$ PFU/g tissue, respectively) (Fig. 7K). The MMM vector control caused severe lung pathology (score of 2.9) whereas the TVC-V and TVC-VI groups had mild to moderate lung pathology (average scores of 1.3 and 1.5,

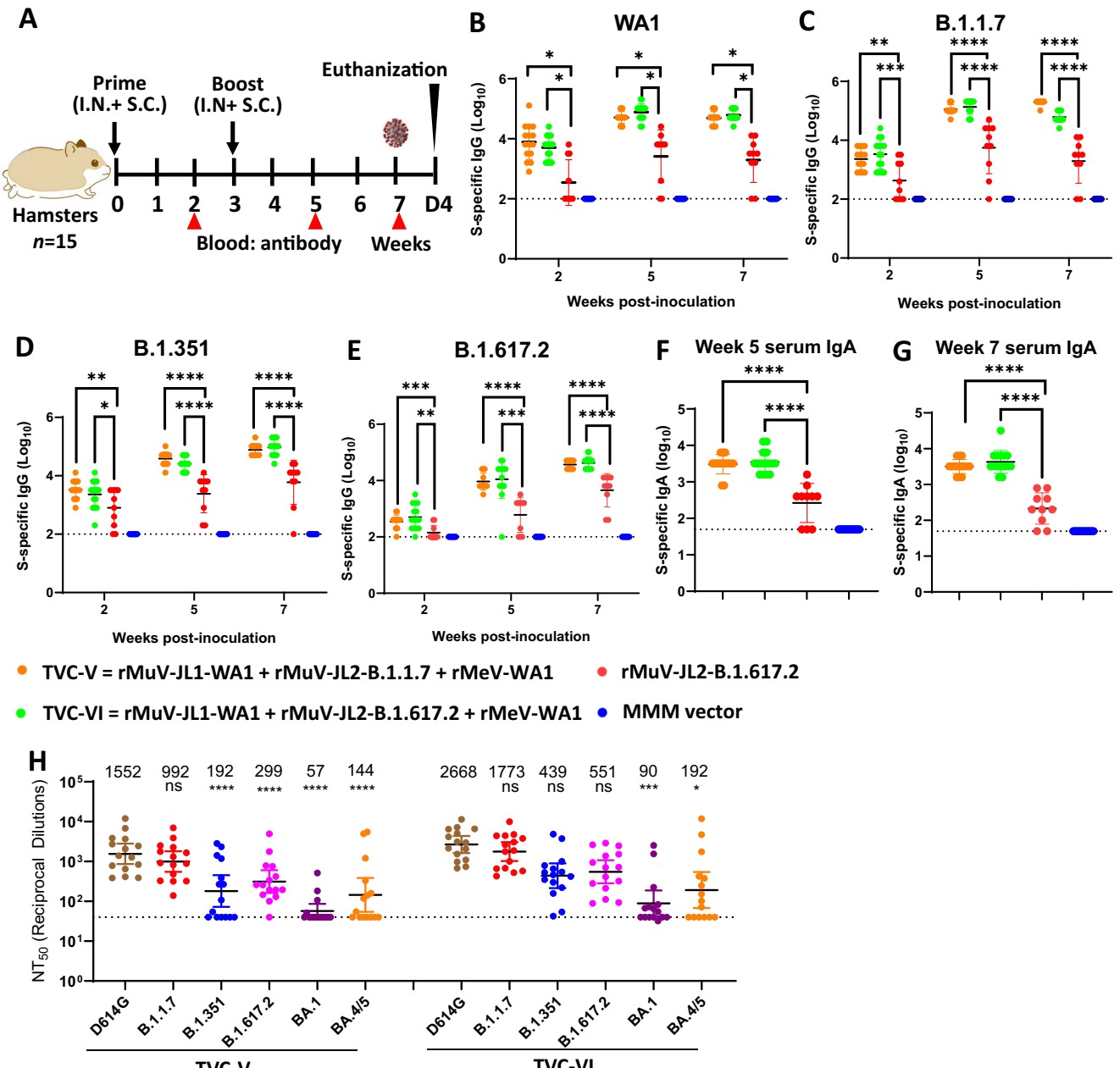

**Fig. 6 | Trivalent vaccine candidates are highly immunogenic in hamsters.**
**A** Immunization schedule in hamsters: female hamsters ($n = 15$) were immunized with $1.2 \times 10^6$ PFU (half S.C. and half I.N.) of TVC-V (rMuV-JL1-WA1, rMuV-JL2-B.1.1.7, and rMeV-WA1), TVC-VI (rMuV-JL1-WA1, rMuV-JL2-B.1.617.2, and rMeV-WA1), rMuV-JL2-B.1.617.2, or MMM vector. Three weeks later, hamsters were boosted with the same vaccine. At weeks 2, 5, and 7, sera ($n = 15$) were collected for detection of S-specific serum IgG antibodies by ELISA using preS-6P of SARS-CoV-2 WA1 (**B**), B.1.1.7 (**C**), B.1.135 (**D**), B.1.617.2 (**E**) as the coating antigen. In addition, S-specific serum IgA titer at weeks 5 (**F**) and 7 (**G**) was determined by ELISA using preS-6P of SARS-CoV-2 WA1 as the coating antigen. **H** Serum NAbs against SARS-CoV-2 VoCs. Sera at week 7 were used for a lentivirus-pseudotyped neutralization assay against SARS-CoV-2 WA1 (D614G), B.1.1.7, B.1.351, B.1.617.2, Omicron BA.1 (B.1.1.529), or BA.4/5 spike. The 50% neutralization titer (NT$_{50}$) was calculated for each serum sample. The mean titers of fifteen hamsters ($n = 15$) ± SD are shown. The limit of

detection is indicated by the dotted line. The *P*-value for TVC-V and TVC-VI vs rMuV-JL2.B.1.617.2 in (**B**–**G**) is: **B**, *$P = 0.0389$ and *$P = 0.0356$ (week 2), *$P = 0.0389$ and *$P = 0.0356$ (week 5), *$P = 0.0389$ and *$P = 0.0356$ (week 7); **C**, **$P = 0.001467$ and ***$P = 0.000811$ (week 2), ****$P = 0.00001$ and ****$P = 0.000007$ (week 5), ****$P = 1.367 \times 10^{-7}$ and ****$P = 1.367 \times 10^{-7}$ (week 7); **D**, **$P = 0.003319$ and *$P = 0.047119$ (week 2), ****$P = 0.000001$ and ****$P = 0.00001$ (week 5), ****$P = 0.000014$ and *$P = 0.00001$ (week 7); **E**, **$P = 0.000627$ and **$P = 0.001518$ (week 2), ****$P = 0.000002$ and **$P = 0.000091$ (week 5), ****$P = 0.000007$ and ****$P = 0.000005$ (week 7); **F**, ****$P = 2.176 \times 10^{-23}$ and ****$P = 2.176 \times 10^{-23}$ (week 5); **G**, ****$P = 2.176 \times 10^{-23}$ and ****$P = 2.176 \times 10^{-23}$ (week 7). In **H**, the *P*-value for B.1.351, B.1.617.2, Omicron BA.1, and BA.4/5 vs WA1(D614G) in TVC-V is ****$P = 3.618 \times 10^{-12}$; Omicron BA.1 and BA.4/5 vs WA1(D614G) in TVC-VI is ****$P = 0.0005$ and ****$P = 0.0228$, respectively. One-way ANOVA was used for statistical analysis (ns > 0.05, *$P < 0.05$; **$P < 0.01$; ***$P < 0.001$; ****$P < 0.0001$).

respectively) (Fig. 7L and Fig. S10). Thus, TVC-V and TVC-VI provide sufficient protection against lung infection but incomplete protection against viral replication in the nose after challenge with the B.1.1.529.

## Parental MeV and MuV vectors do not have adjuvant effects on the SARS-CoV-2 immune response
We compared the SARS-CoV-2-specific immune responses between rMuV-JL2-WA1 alone and rMuV-JL2-WA1 with vector viruses (null

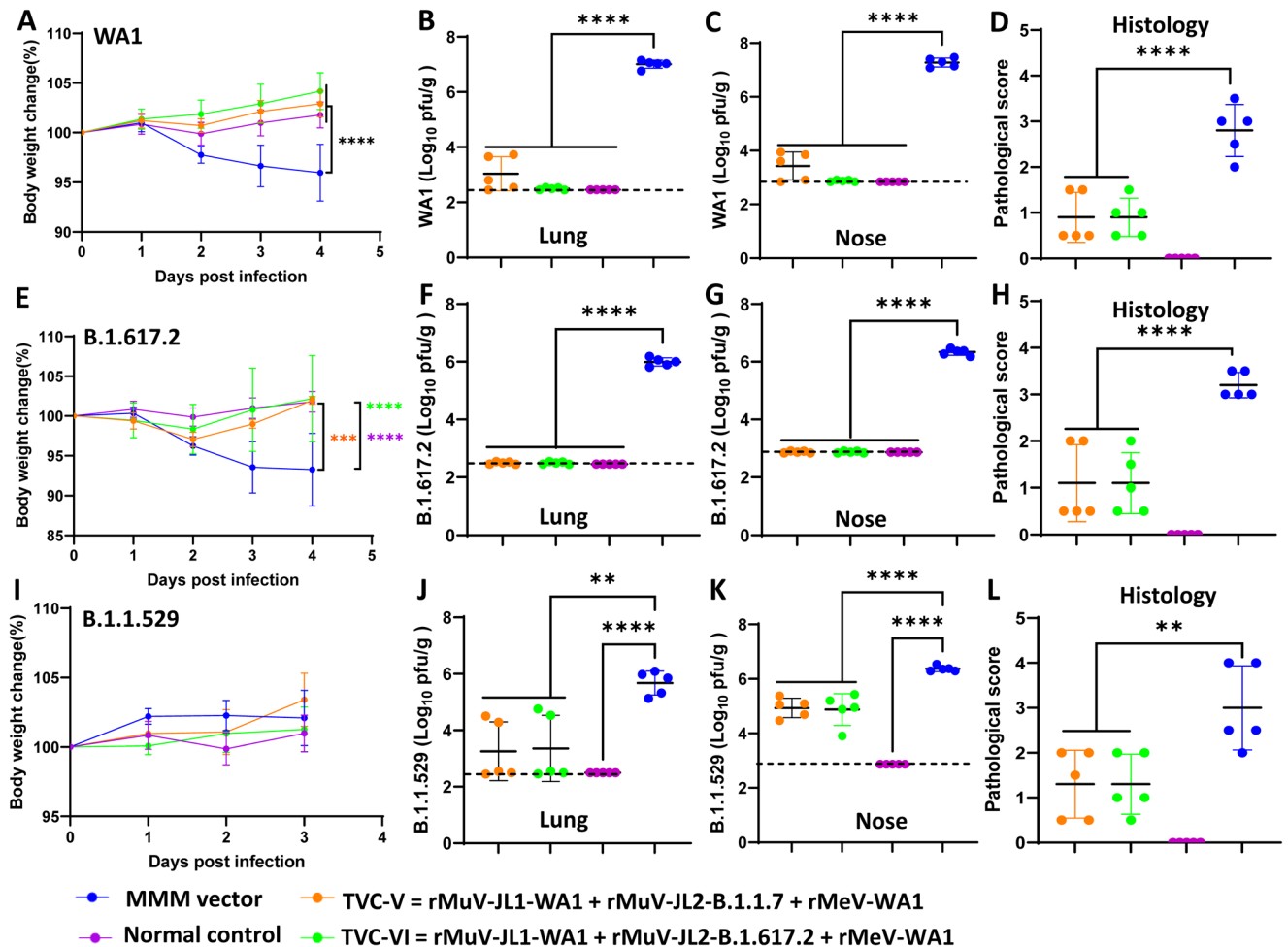

**Fig. 7 | Trivalent vaccine candidates (TVC-V and TVC-VI) protect hamsters against challenge with SARS-CoV-2 WA1 and variants of concern.** At week 7, 5 hamsters ($n = 5$) in each group in Fig. 6A were challenged with SARS-CoV-2 WA1 (**A–D**), B.1.617.2 (**E–H**), or B.1.1.529 (**I–L**). Changes in body weight after challenge with SARS-CoV-2 WA1 (**A**), B.1.617.2 (**E**) or B.1.1.529 (**I**). Normal refers to animals inoculated with DMEM. Percent of weight on the challenge day was shown. Data are the average of 5 hamsters ($n = 5$) ± SD. Viral burden in the lung after challenge with SARS-CoV-2 WA1 (**B**), B.1.617.2 (**F**), or B.1.1.529 (**J**). Viral burden in the nasal turbinate after challenge with SARS-CoV-2 WA1 (**C**), B.1.617.2 (**G**), or B.1.1.529 (**K**). Hamsters challenged with SARS-CoV-2 WA1 or B.1.617.2 were euthanized at day 4 whereas hamsters challenged with B.1.1.529 were euthanized at day 3. Lung histopathological score after challenge with SARS-CoV-2 WA1 (**D**), B.1.617.2 (**H**), or B.1.1.529 (**L**). Viral titers are the GMT of 5 hamsters ± SD. The dotted line indicates the detection limit. Pathology of each lung section was scored based on the severity of histologic changes. Score of 0, 1, 2, 3, and 4 represents no, mild, moderate, severe, and extremely severe pathological changes. Data are the mean of 5 hamsters ± SD. The *P*-value for TVC-V, TVC-VI, and normal vs MMM vector is: **A**, ****$P < 1 \times 10^{-15}$, ****$P < 1 \times 10^{-15}$, and ****$P = 2.974 \times 10^{-8}$; **B**, ****$P = 6.38 \times 10^{-12}$, ****$P = 1.08 \times 10^{-12}$, and ****$P = 9.53 \times 10^{-13}$; **C**, ****$P = 1.34 \times 10^{-12}$, ****$P = 1.6 \times 10^{-13}$, and ****$P = 1.5 \times 10^{-13}$; **D**, ****$P < 1 \times 10^{-15}$, ****$P < 1 \times 10^{-15}$, and ****$P < 1 \times 10^{-15}$; **E**, ***$P = 4.19 \times 10^{-4}$, ****$P = 9.59 \times 10^{-6}$, and ****$P = 3.37 \times 10^{-7}$; **F**, **$P = 2.3 \times 10^{-14}$, ****$P = 2.3 \times 10^{-14}$, and ****$P = 2.3 \times 10^{-14}$; **G**, *** $P = 2.3 \times 10^{-14}$, **** $P = 2.3 \times 10^{-14}$, and **** $P = 2.3 \times 10^{-14}$; **H**, ****$P < 1 \times 10^{-15}$, **** $P < 1 \times 10^{-15}$, and **** $P < 1 \times 10^{-15}$; **I**, $P = 0.8766$, *$P = 0.028$, and ****$P = 0.0086$; **J**, **$P = 0.0012$, **$P = 0.0018$, and ****$P = 7.208 \times 10^{-5}$; **K**, ****$P = 3.505 \times 10^{-5}$, ****$P = 2.170 \times 10^{-5}$, and ****$P = 1.575 \times 10^{-10}$; **L**, **$P = 0.0035$, **$P = 0.0035$, and **** $P < 1 \times 10^{-15}$. Data were analyzed using two-way ANOVA and one-way ANOVA (*$P < 0.05$; **$P < 0.01$; ***$P < 0.001$; ****$P < 0.0001$).

rMeV + null rMuV-JL1). At week 2, the rMuV-JL2-WA1 + null rMeV + null rMuV-JL1 group had significantly lower IgG than the rMuV-JL2-WA1 group ($P = 2.04 \times 10^{-5}$) (Fig. S11). However, there was no significant difference in serum IgG between the two groups at weeks 5 ($P = 0.1556$) and 7 ($P = 0.2842$) (Fig. S11). Thus, the addition of the null co-virus to MuV-JL2-WA1 does not significantly interfere with SARS-CoV-2 antibody responses.

## A low dose of TVC is sufficient to induce a strong immune response

We compared the effects of doses on immune responses of trivalent vaccines. We combined rMuV-JL2-WA1+rMuV-JL1-WA1+rMeV-WA1 to generate TVC-VII (3 viruses expressing a single preS-6P of SARS-CoV-2 WA1) and combined rMeV-WA1+rMuV-JL1-B.1.617.2+rMuV-JL2-B.1.1.7 to generate TVC-VIII (3 viruses expressing three different preS-6P

proteins) (Table S2). In parallel, we compared the low ($3 \times 10^4$ PFU), medium ($3 \times 10^5$ PFU), and high dose ($1.2 \times 10^6$ PFU) of TVC-VII (Fig. 8A–F) and TVC-VIII (Fig. 8G–L) in hamsters. We found that all three doses induced strong serum IgG, serum IgA, and NAbs. For serum IgG (Fig. 8A, B, C, G, H, and I) and IgA (Fig. 8D, E, J, and K), no significant difference in WA1-, B.1.617.2-, or B.1.1.7-specific IgG titers was observed between medium and high doses at three time points (weeks 2, 5, and 7). However, the IgG and IgA titers in the low dose were significantly lower than the high dose at most time points. Next, week 7 sera from low and high dose groups were chosen to examine NAbs. For both TVC-VII and TVC-VIII, high dose group induced higher NAb than the low dose group although there was no significant difference between low and high doses against all variants (Fig. 8F and L). Thus, a low dose ($3 \times 10^4$ PFU) of trivalent vaccines is sufficient for inducing strong SARS-CoV-2-specific NAbs.

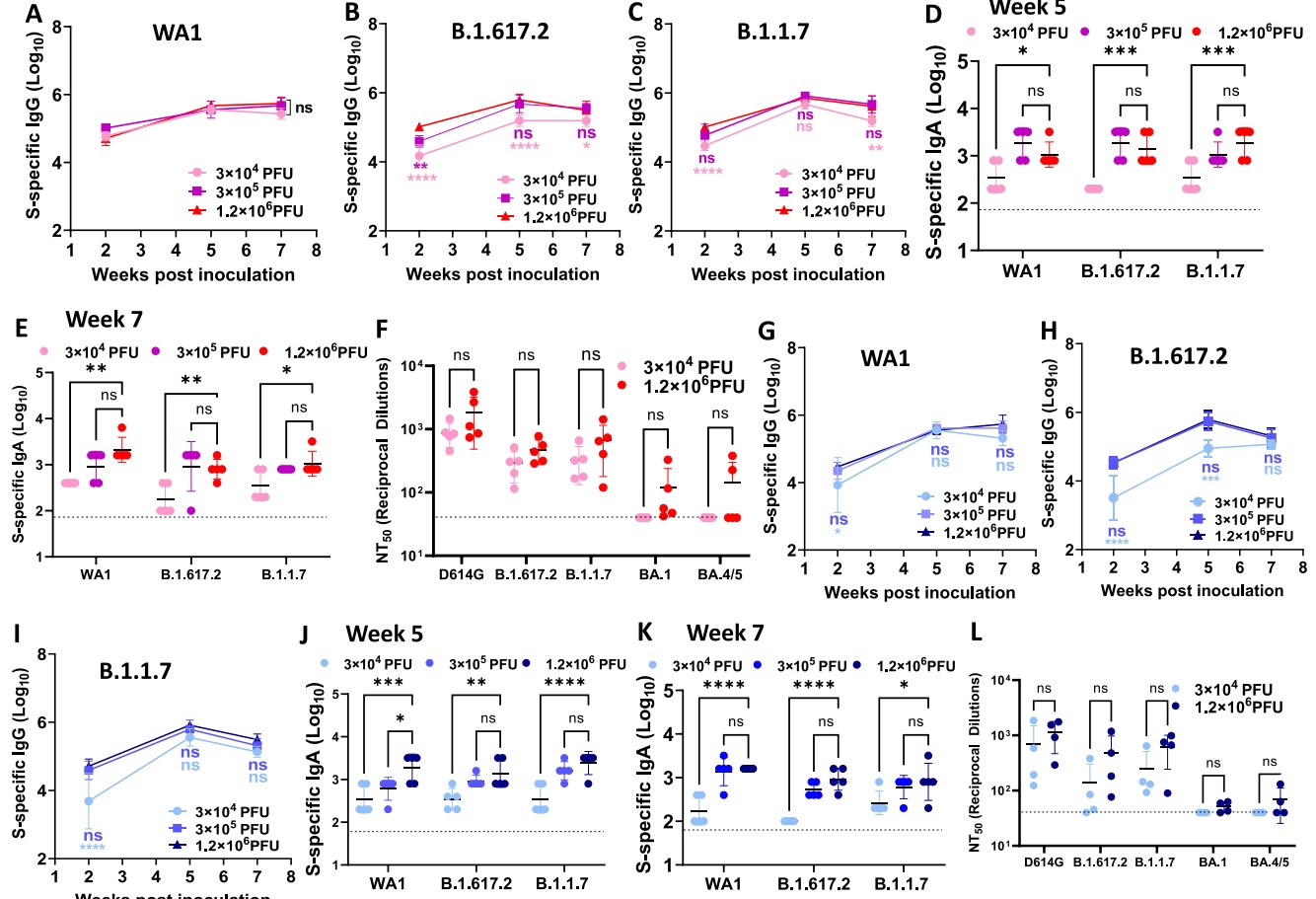

**Fig. 8 | The effects of doses and antigen compositions on the immune responses of trivalent vaccines.** 30 female hamsters were randomly divided into 6 groups ($n = 5$). The first 3 groups (**A**–**F**) were immunized I.N. with $3 \times 10^4$, $3 \times 10^5$, and $1.2 \times 10^6$ PFU of TVC-VII (rMuV-JL2-WA1 + rMuV-JL1-WA1 + rMeV-WA1) and the other 3 groups (**G**–**L**) were immunized I.N. with $3 \times 10^4$, $3 \times 10^5$, and $1.2 \times 10^6$ PFU of TVC-VIII (rMeV-WA1+rMuV-JL1-B.1.617.2+rMuV-JL2-B.1.1.7). Three weeks later, each group was boosted I.N. with the same dose of the same vaccine. WA1 (**A**, **G**), B.1.617.2 (**B**, **H**), and B.1.1.7 (**C**, **I**) specific serum IgG titers were determined by ELISA. Week 5 (**D**, **J**) and week 7 (**E**, **K**) serum IgA titers were determined by ELISA. Week 7 sera were used for the determination of NAbs (**F**, **L**) against different VoCs by pseudotyped virus neutralization assay. IgG and IgA titers are the GMT of 5 hamsters ($n = 5$) ± SD. NAb titers are the mean titers of five hamsters ($n = 5$) ± SD. In **B**, the $P$-value for $3 \times 10^4$ PFU vs $1.2 \times 10^6$ PFU at weeks 2, 5, and 7 is ****$P < 1.0 \times 10^{-15}$, ****$P < 1.0 \times 10^{-15}$, and *$P = 0.0457$, respectively. In **C**, the $P$-value for $3 \times 10^4$ PFU vs

$1.2 \times 10^6$ PFU at weeks 2 and 7 is ***$P = 0.0002$ and **$P = 0.0031$, respectively. In **D**, the $P$-value for $3 \times 10^4$ PFU vs $1.2 \times 10^6$ PFU at weeks 2, 5, and 7 is *$P = 0.0277$, ***$P = 0.000147$, and ***$P = 0.000966$, respectively. In **E**, the $P$-value for $3 \times 10^4$ PFU vs $1.2 \times 10^6$ PFU at weeks 2, 5, and 7 is **$P = 0.00103$, **$P = 0.0024$, and *$P = 0.0287$, respectively. In **G**, the $P$-value for $3 \times 10^4$ PFU vs $1.2 \times 10^6$ PFU at weeks 2 is *$P = 0.026$. In **H**, the $P$-value for $3 \times 10^4$ PFU vs $1.2 \times 10^6$ PFU at weeks 2 and 5 is ****$P < 1.0 \times 10^{-15}$ and ***$P = 0.000147$, respectively. In **I**, the $P$-value for $3 \times 10^4$ PFU vs $1.2 \times 10^6$ PFU at weeks 2 is ****$P < 1.0 \times 10^{-15}$. In **J**, the $P$-value for $3 \times 10^4$ PFU vs $1.2 \times 10^6$ PFU at weeks 2, 5, and 7 is ***$P = 0.0005$, **$P = 0.0032$, and ****$P < 1.0 \times 10^{-15}$, respectively. In **K**, the $P$-value for $3 \times 10^4$ PFU vs $1.2 \times 10^6$ PFU in WA1, B.1.617.2, and B.1.1.7 is ****$P < 1.0 \times 10^{-15}$, ****$P < 1.0 \times 10^{-15}$, and *$P = 0.013$, respectively. Statistical analyses were conducted using one-way (**D**–**F** and **J**–**L**) or two-way (**A**–**C** and **G**–**I**) ANOVA (*$P < 0.05$; **$P < 0.01$; ***$P < 0.001$; ****$P < 0.0001$; ns not significant).

## Trivalent vaccines expressing three spikes of VoCs or WA1 spike induces similar levels of antibodies

We re-analyzed the data in Fig. 8 by comparing the WA1(D614G)-, B.1.617.2-, and B.1.1.7-specific serum IgG, IgA, and NAb titer between TVC-VII and TVC-VIII. This analysis showed no significant difference in WA1(D614G)-, B.1.617.2-, and B.1.1.7-specific serum IgG or IgA titers between TVC-VII and TVC-VIII at low (Fig. S12), medium (Fig. S13), and high (Fig. S14) doses. In addition, there was no significant difference in WA1(D614G)-, B.1.617.2-, and B.1.1.7-specific NAb titers between TVC-VII and TVC-VIII at low (Fig. S15A) or high (Fig. S15B) doses. Thus, TVC-VII and TVC-VIII induce similar immune responses against WA1(D614G), B.1.617.2, and B.1.1.7.

## The preexisting MMM vector immunity does not significantly interfere with SARS-CoV-2-specific antibodies

Hamsters in Group 1 were first immunized S.C. with MMM vector to induce the preexisting immunity and the hamsters in Group

2 served as controls with no preexisting immunity (Fig. 9A). At week 2, MuV and MeV-specific NAbs were observed in the MMM vector (Group 1) but not in the DMEM control (Group 2). MuV and MeV-specific NAbs in Group 1 were higher than those in Group 2 in the following weeks (Fig. 9B, C), demonstrating that the pre-existing NAb against MuV and MeV has been induced. At weeks 3 and 5, both groups were immunized I.N. with TVC-VIII (rMeV-WA1 + rMuV-JL1-B.1.617.2 + rMuV-JL2-B.1.1.7). At week 5, Group 1 had ~8 times lower serum IgG specific for WA1 (Fig. 9D), B.1.617.2 (Fig. 9E), and B.1.1.7 (Fig. 9F) than Group 2. At week 7, the Group 1 had ~1.5 times lower WA1, B.1.617.2, and B.1.1.7-specific serum IgG than Group 2. However, at weeks 9 and 11, there was no significant difference in SARS-CoV-2-specific IgG titer between Groups 1 and 2 (Fig. 9D–F). Thus, SARS-CoV-2 specific antibody response is delayed at weeks 5 and 7 in the presence of pre-existing MMM vector immunity but, nevertheless, reached similar titers by weeks 9 and 11.

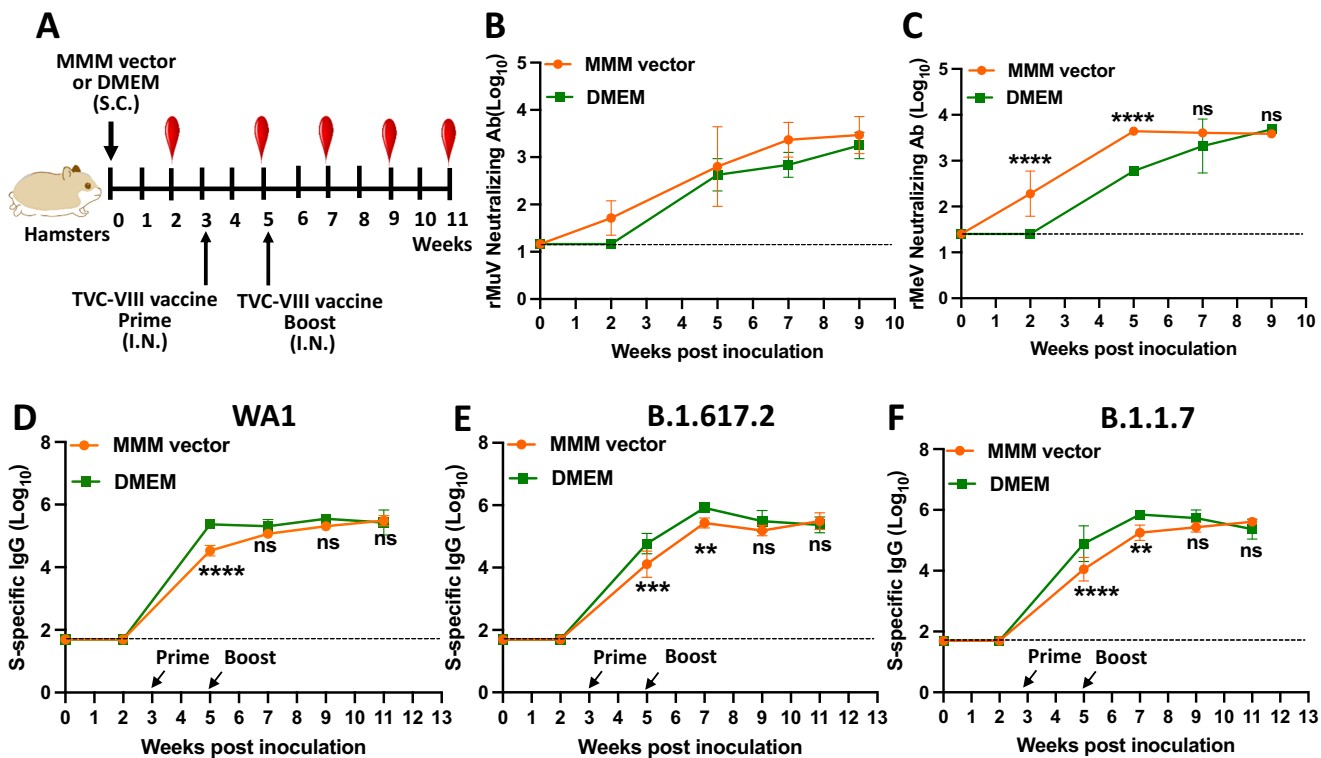

**Fig. 9 | The preexisting MMM vector immunity does not significantly interfere with SARS-CoV-2-specific antibodies induced by a trivalent vaccine.** **A** Immunization schedule. Two groups of female hamsters (*n* = 5) were inoculated S.C. with 1.2 × 10⁶ PFU of MMM vectors (a mixture of rMeV, rMuV-JL1, and rMuV-JL2, 4 × 10⁵ PFU per virus) or DMEM. Three weeks later, both groups were immunized with 1.2 × 10⁶ PFU of TVC-VIII (a mixture of rMeV-WA1 + rMuV-JL1-B.1.617.2 + rMuV-JL2-B.1.1.7, 4 × 10⁵ PFU per virus). At week 5, both groups were boosted with 1.2 × 10⁶ PFU of TVC-VIII. **B** MuV-specific NAb response measured by plaque reduction neutralization assay. **C** MeV-specific NAb response measured by plaque reduction neutralization assay. The *P*-value at weeks 2 and 5 is ****P = 7.874 × 10⁻⁶ and ****P = 1.096 × 10⁻⁵, respectively. **D–F** WA1 (**D**), B.1.617.2 (**E**), and B.1.1.7 (**F**) -specific serum IgG titers were measured by ELISA. In **E**, the *P*-value at weeks 5 and 7 is ***P = 0.000138 and **P = 0.0079, respectively. In **F**, the *P*-value at weeks 5 and 7 is ****P = 2.444 × 10⁻⁵ and **P = 0.0032, respectively. MeV NAb, MuV NAb, and IgG titers are the GMT of 5 hamsters (*n* = 5) ± SD. Two-way ANOVA was used for statistical analysis (ns > 0.05; **P < 0.01; ****P < 0.0001).

## The effect of immune imprinting on the efficacy of the trivalent vaccine

Hamsters in Groups 1-3 were first received two doses of rMuV-JL2-WA1 at weeks 0 and 3 to induce a strong WA1-specific immunity (Fig. 10A), followed by immunization with the third dose of trivalent TVC-IX (rMeV-BA.1, rMuV-JL1-B.1.617.2, and rMuV-JL2-WA1) (Group 1), monovalent rMeV-BA1 (Group 2), and monovalent rMuV-JL2-WA1 (Group 3) vaccine at week 5, respectively. At week 7, hamsters receiving the third booster of monovalent rMeV-BA.1 and monovalent rMuV-JL2-WA1 exhibited significantly lower WA1-specific IgG against B.1.617.2- and BA.1-specific IgG whereas hamsters receiving the third booster of trivalent vaccine had a similar level of serum IgG against WA1, B.1.617.2, and BA.1 spikes (Fig. 10B). At week 9, serum IgG antibodies reached a similar level against WA1, B.1.617.2, and BA.1 spikes in Groups 1, 2, and 3 (Fig. 10C). Subsequently, week 9 sera were used in a neutralization assay (Fig. 10D and Fig. S16). In Group 1, WA1 (D614G)-specific NAb titer is the highest, B.1.617.2-specific NAb is the second, and BA.1-specific NAb titer is the lowest, although there was no significant difference among them. A similar trend was observed in Groups 2 and 3. In Group 2, WA1 (D614G)-specific NAb titer was significantly higher than B.1.617.2- and BA.1-specific NAb titers. In addition, B.1.617.2-specific NAb titer was significantly higher than BA.1-specific NAb. In Group 3, WA1 (D614G)-specific NAb titer was similar to B.1.617.2-specifc antibody but was significantly higher than BA.1-specific NAb. In addition, Group 2 (monovalent rMeV-BA.1) induces 4.0, 3.2, and 2.5-fold higher D614G-, B.1.617.2-, and BA.1-specific NAb than Group 1 (trivalent vaccine), respectively (Fig. S16C). Thus, monovalent BA.1 vaccine is more effective in inducing BA.1-specfic NAb than the trivalent vaccine,

indicating that immune imprinting reduces the BA.1-specific antibody response.

Finally, we determined whether Groups 1–3 were protected from Omicron BA.1 challenge. None of the challenged groups had significant weight loss (Fig. 10E). At day 3 post-challenge, BA.1 viral titer in the lungs of Groups 1–3 was below or near the detection limit whereas challenge control had 5 log₁₀ PFU/g tissue of viral titer in lungs (Fig. 10F). In addition, BA.1 viral titer in nasal turbinates of Groups 1–2 was near the detection limit whereas Group 3 had ~4 log₁₀ PFU/g tissue (Fig. 10G). Thus, all three groups were provided with near complete protection against an Omicron BA.1 challenge.

## Discussion

In this study, we generated rMeV, rMuV-JL1, and rMuV-JL2 expressing preS-6P proteins of original SARS-CoV-2 WA1 and several VoCs (B.1.1.7, B.1.351, B.1.617.2, and B.1.1.529), and tested the immunogenicity of several TVC combinations. We found that TVC: (i) induced higher serum antibody responses than the monovalent vaccine candidate; (ii) intranasal delivery was the most effective immunization route for inducing systemic and mucosal antibody responses, and Trm cell responses; (iii) a medium (total of 3 × 10⁵ PFU) dose of a trivalent vaccine induces levels of antibody and T cell immune responses similar to a high (total of 1.2 × 10⁶ PFU) dose in mice; (iv) a low dose (total of 3 × 10⁴ PFU) of TVC induces a similar level of NAb compared to a high (total of 1.2 × 10⁶ PFU) dose in hamsters; (v) a trivalent vaccine containing preS-6P proteins of WA1, B.1.1.7, and B.1.617.2 provides complete protection against challenge with WA1 and B.1.617.2 in hamsters; (vi) a trivalent vaccine containing preS-6P of WA1, B.1.1.7, and B.1.617.2

Group 1: rMuV-JL2-WA1 (1st immunization) + rMuV-JL2-WA1 (2nd immunization) + TVC-IX (3rd immunization)
Group 2: rMuV-JL2-WA1 (1st immunization) + rMuV-JL2-WA1 (2nd immunization) + monovalent rMeV-BA.1 (3rd immunization)
Group 3: rMuV-JL2-WA1 (1st immunization) + rMuV-JL2-WA1 (2nd immunization) + rMuV-JL2-WA1 (3rd immunization)
Group 4: rMuV-JL2 (1st immunization) + rMuV-JL2 (2nd immunization) + rMuV-JL2 (3rd immunization)

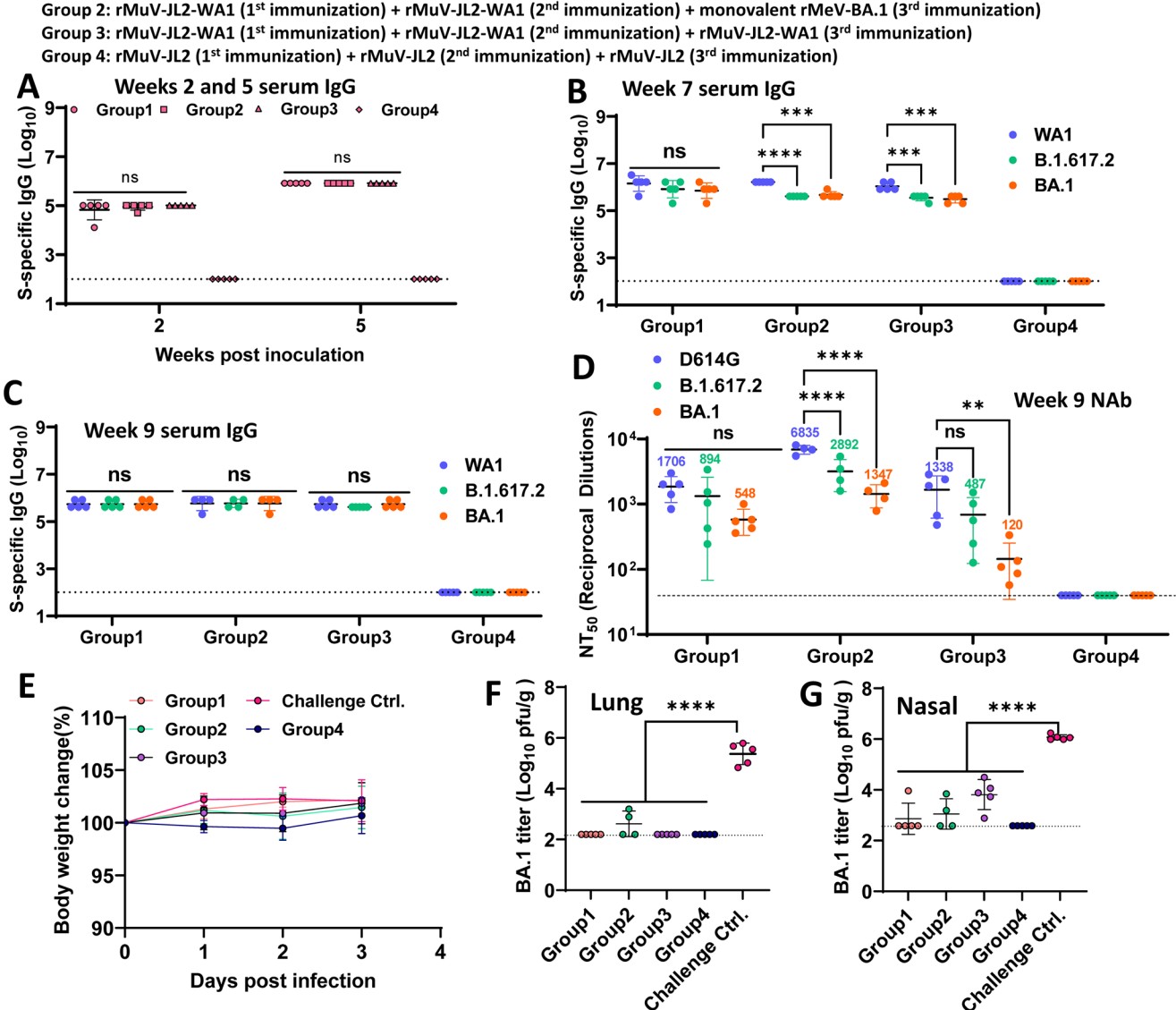

**Fig. 10 | The impact of immune imprinting on the efficacy of trivalent vaccines.** 20 female hamsters were divided into 4 groups ($n = 5$). Groups 1, 2, and 3 received two doses of rMuV-JL2-WA1 ($1.2 \times 10^6$ PFU) at weeks 0 and 3, and Group 4 received two doses of rMuV-JL2 control. At week 5, Groups 1, 2, and 3 received $1.2 \times 10^6$ PFU of trivalent TVC IX (rMeV-BA.1, rMuV-JL1-B.1.617.2, and rMuV-JL2-WA1), monovalent rMeV-BA.1, and monovalent rMuV-JL2-WA1, respectively. **A** WA1-specific IgG at weeks 2 and 5. **B** WA1-, B.1.617.2-, and BA.1-specific IgG at week 7. The *P*-value for WA1 vs. BA.1 in Group 1 is *$P = 0.0439$. The *P*-value for B.1.617.2 and BA.1 vs WA1 in Group 2 is ****$P = 2.8251 \times 10^{-5}$ and ***$P = 0.000147$, respectively. The *P*-value for B.1.617.2 and BA.1 vs WA1 in Group 3 is ***$P = 0.0007$ and ***$P = 0.00015$, respectively. **C** WA1-, B.1.617.2-, and BA.1-specific IgG at week 9. IgG titers are the GMT of 5 hamsters ($n = 5$) ± SD. **D** WA1(D614G)-, B.1.617.2-, and BA.1-specific NAb at week 9. NAb was detected by pseudotype neutralization assay. NAb titers are the mean of 5 hamsters ± SD are shown. The *P*-value for B.1.617.2 and BA.1 vs WA1 in Group 2 is ****$P = 8.197 \times 10^{-8}$ and ****$P = 2.3 \times 10^{-12}$, respectively. The *P*-value for BA.1 vs WA1 in Group 3 is **$P = 0.0095$. **E** Body weight changes in hamsters after Omicron BA.1 challenge. Percent of weight at the challenge day was shown. Data are the average of 5 hamsters ($n = 5$) ± SD. **F** Omicron BA.1 titer in lung of hamsters after BA.1 challenge. The *P*-value for Group 1, 2, 3, and 3 vs challenge control is ****$P < 1.0 \times 10^{-15}$, ****$P < 1.0 \times 10^{-15}$, ****$P < 1.0 \times 10^{-15}$, and ****$P < 1.0 \times 10^{-15}$, respectively. **G** Omicron BA.1 titer in nasal turbinate of hamsters after BA.1 challenge. Viral titers are the GMT of 5 hamsters ($n = 5$) ± SD. The dotted line indicates the detection limit. The *P*-value for Group 1, 2, 3, and 3 vs challenge control ****$P < 1.0 \times 10^{-15}$, ****$P < 1.0 \times 10^{-15}$, ****$P < 1.0 \times 10^{-15}$, and ****$P < 1.0 \times 10^{-15}$, respectively. Data were analyzed using two-way ANOVA and one-way ANOVA (ns > 0.05; **$P < 0.01$; ***$P < 0.001$; ****$P < 0.0001$).

induces substantial protection against challenge with Omicron BA.1 in hamsters despite the lack of high levels of Omicron BA.1-specific NABs; (vii) preexisting MMM vector immunity delays slightly but does not significantly reduce the SARS-CoV-2-specific antibody induced by the trivalent vaccine; and (viii) immune imprinting induced by previous WA1 preS-6P immunization reduces the Omicron BA.1-specific antibody responses of trivalent vaccine.

The MMR vaccine is one of the most successful vaccines in human history. It has a high safety profile and elicits long-term or lifelong protection against three major childhood viral diseases (measles,

mumps, and rubella)[24,42]. Our SARS-CoV-2 TVC, which utilize the Edmonston vaccine strain of MeV as well as the major component MuV-JL2 and the minor component MuV-JL1 strains that comprise the MMR vaccine, can incorporate three preS-6P proteins from multiple variants within each vector. Although the primary target of both MeV and MuV is the respiratory tract, they utilize different receptors, have different host tropisms, infecting different tissues and cell types[24,42]. Therefore, combination of MeV, MuV-JL1, and MuV-JL2 expressing different preS-6P proteins or same preS-6P may offer synergistic effects, providing broader protection against these VoCs and their relatives.

We found that the preexisting MeV and MuV antibodies had a minimal impact on the efficacy of trivalent vaccines in a hamster model. SARS-CoV-2-specific antibody had a delay in weeks 5 and 7 but reached a similar titer at weeks 9 and 11 in the presence of the pre-existing MMM vector immunity. Our previous study showed that the pre-existing MuV antibody did not significantly interfere with the SARS-CoV-2-specific antibody induced by rMuV-preS-6P in mice[32]. Several MeV-based vaccine candidates (HIV-1, Zika virus, and chikungunya virus) have successfully completed phase I clinical trials in adults[24,42,43]. The MeV-based chikungunya vaccine has been shown to be safe and highly immunogenic in humans in the presence of pre-existing anti-MeV immunity[42].

Interestingly, we found that Omicron BA.1-specfic NAbs induced by TVC-V and TVC-VI are very low, however, both trivalent vaccines can provide substantial protection against the challenge with Omicron BA.1 in hamsters. This includes a near detection limit level of viral titer in the lungs, a significant reduction of lung pathology, and a significant reduction of viral titers in nasal turbinates, suggesting that T cell responses, to some extent, contribute to the protection against Omicron BA.1. In fact, several studies have demonstrated that T cell responses, unlike NAb responses, are highly cross-reactive against multiple SARS-CoV-2 variants, including Omicron BA.1[44–46].

In the US, the MMR vaccine is generally administered subcutaneously to infants and children, and intramuscularly in some countries[20]. Interestingly, human clinical trials of individual MeV or MuV vaccines have found that intranasal vaccination is more efficacious than subcutaneous in inducing MeV or MuV-specific antibodies[20,22,27,47]. We compared the efficacy of intranasal, subcutaneous, and a combination of intranasal and subcutaneous immunization. Interestingly, the intranasal route was found to be the most effective immunization route in inducing both serum NAbs and mucosal immunity. The combination intranasal and subcutaneous routes was the second best whereas subcutaneous ranked last.

We found that both intranasal and subcutaneous routes of our TVC induce strong Th1/17-biased cellular immune responses in the spleen. However, the intranasal but not the subcutaneous route induced local lung IgA antibodies as well as serum IgA. Importantly, intranasal immunization induced strong S-specific Trm in the lungs, a signature of an effective intranasal vaccine[48]. Unlike circulating memory T cell populations that patrol blood and lymph, Trm are the predominant surveyors of nonlymphoid tissues and accelerate pathogen control in the event of local infection[40,48]. They can rapidly respond to pathogens through direct release of cytotoxic mediators, cytokines, and chemokines[40,48]. For example, IFN-γ has been shown to broadly enhance tissue-wide antiviral responses such as upregulating the type I IFN signaling pathway factors and the enhancing leukocyte recruitment to the site of infection[49]. In addition to direct production of effector cytokines, Trm proliferate in situ in response to locally encountered antigen and interact with other immune cells, both within the tissue, and promote enhanced communication with local lymphoid sites[50]. Therefore, the induction of Trm by intranasal immunization of trivalent vaccine is one of the major advantages of intranasal delivery because Trm in the lung is important for site-specific protection. The induction of serum NAbs, systemic immune T cells, lung IgA, and Trm will likely enhance the ability of SARS-CoV-2 vaccines to protect against not only severe disease but also viral transmission.

Antigenic distance between variants is critical for justifying the development of a variant-specific vaccines and must be considered when designing a trivalent vaccine regimen. We found that a trivalent vaccine expressing the B.1.1.7, B.1.617.2, and WA1 spikes induced a level of B.1.1.7-, B.1.617.2-, and WA1-specific NAb similar to a trivalent vaccine containing the WA1 spike, alone. Because B.1.1.7 and B.1.617.2 are antigenically similar to the SARS-CoV-2 WA1, the benefit of combining B.1.1.7, B.1.617.2, and WA1 spikes in a trivalent vaccine may be limited. Importantly, we recently found that a trivalent vaccine containing BA.1,

B.1.617.2, and WA1 spikes generated broad NAb against WA1, B.1.617.2, and Omicron BA.1 and BA4/5 VoCs whereas the monovalent vaccine containing BA.1 spike induced antibodies that only neutralize the homologous BA.1 virus, not the heterologous viruses[51]. Because Omicron BA.1 is antigenically far from WA1 and B.1.617.2, a trivalent vaccine that contains the spike of BA.1 may be capable of inducing more broadly NAbs. Thus, the value of including a VoC spike in a trivalent vaccine will likely be dependent on the antigenic distance between the VoCs, or between the VoC and the original SARS-CoV-2 WA1.

Immune imprinting is another critical factor which should be considered in vaccine antigen design[52,53]. To prevent Omicron BA.4/5 infection, Pfizer and Moderna developed the SARS-CoV-2 bivalent vaccine expressing the WA1 and Omicron BA.4/5 spike proteins. Unfortunately, the immune imprinting caused by the previous WA1 spike protein vaccination boosted the production of antibodies against WA1 more than inducing antibodies against the novel Omicron BA4/5 epitopes, significantly compromising the specific Omicron antibody response to the bivalent vaccine booster[54,55]. To reduce the immune imprinting, the FDA recently approved a monovalent XBB.1.5 mRNA vaccine rather the bivalent mRNA vaccine to prevent Omicron XBB.1.5[14,56]. In this study, we found that monovalent rMeV-BA.1 vaccine induced 2.5-fold higher BA.1-specific NAb than the trivalent vaccine in the presence of the WA1-neutralizing antibody, indicating our trivalent vaccine approach may be also impacted by immune imprinting. However, it should be noted that the dose of rMeV-BA.1 in the trivalent vaccine is 3 times less than in the monovalent rMeV-BA.1 vaccine, which may also contribute to the lower BA.1-specific antibody induced by our trivalent vaccine.

There are several limitations in this study. First, the current study did not use the spikes of the currently dominant Omicron subvariants (e.g. JN.1). SARS-CoV-2 has continued to evolve rapidly. At the time of submission of this manuscript, Alpha, Beta, and Delta VoCs had been replaced by Omicron variants and subvariants. Second, antigenic homology should be considered in design of the trivalent vaccine platform. The trivalent vaccine approach is most attractive for presenting multiple antigenically distinct spike variants (such as those from Omicron HV.1, XBB.1.5, and JN.1) enabling the induction of broader NAbs, and for presenting T cell antigens (such as N protein) to induce T cell responses to T cell epitopes. Third, like other COVID-19 vaccines, the trivalent vaccine approach may encounter issues with immune imprinting. At this stage, most of the population has been infected or immunized with the spike derived from the original SARS-CoV-2, or both. To reduce the problems caused by immune imprinting, the WA1 spike should be eliminated from the trivalent vaccine design. Fourth, the immunization doses used this study are higher than those used to vaccinate humans. However, we demonstrate that a relatively low dose of $3 \times 10^4$ PFU is sufficient to induce a strong immune response in hamsters. Future studies will determine the minimal dose required for a strong immune response in rodent models. Finally, IFNAR1$^{-/-}$ mice which are defective innate immune responses were used in this study because immunocompetent mice are not susceptible to MeV[37] or MuV[38] infection. However, hamsters have been effectively used as another small animal model to validate the immune responses and protection in an immune competent animal.

In summary, we have developed multiple intranasal MeV-MuV-based trivalent vaccines, each expressing three distinct SARS-CoV-2 preS-6P proteins. This vaccine platform can be rapidly updated to include current circulating Omicron subvariants HV.1, XBB.1.5, EG.5, and JN.1.

## Methods

### Cells lines and SARS-CoV-2 virus stocks

Vero CCL81 cells (African green monkey, ATCC no. CCL81), Vero E6 cells (ATCC CRL-1586), and HEp-2 cells (ATCC no. CCL-23) were purchased from ATCC (Manassas, VA, USA) and were grown at 37 °C in

Dulbecco's modified Eagle's medium (DMEM; Life Technologies, Carlsbad, CA, USA) supplemented with 10% fetal bovine serum (FBS). FreeStyle™ 293-F cells (Catalog no. R79007) were purchased from ThermoFisher Scientific (Waltham, MA, USA). The SARS-CoV-2 USA-WA1/2020 (WA1) natural isolate (NR-52281, accession no. MN985325), mouse-adapted (MA) SARS-CoV-2 USA-WA1/2020 (strain MA10) (NR-55329, accession no. MT952602), SARS-CoV-2 Delta (B.1.617.2) (NR-55672, GISAID: EPI_ISL_2331496), and SARS-CoV-2 Omicron BA.1 (B.1.1.529) (NR-56461, GISAID: EPI_ISL_7160424) were obtained from BEI Resources (Manassas, VA, USA).

## Construction of MeV and MuV plasmids
The plasmids encoding the full-length genomic cDNA of MuV (JL1 or JL2) or MeV Edmonston vaccine strain with inserted SARS-CoV-2 pre-fusion S with six prolines gene (*preS-6P*) were constructed using a yeast-based recombination system described previously[32,34,35]. The *preS-6P* gene was flanked by the MuV or MeV gene start and gene end sequences and inserted into the P-M junction. Using this method, the *preS-6P* gene of SARS-CoV-2 WA1, B.1.1.7 (Alpha), B.1.351 (Beta), and B.1.617.2 (Delta) was inserted into the genome of MuV-JL2 strain or MeV Edmonston strain, which generated four MuV-JL2 plasmids (pMuV-JL2-WA1, pMuV-JL2-B.1.351, pMuV-JL2-B.1.1.7, and pMuV-JL2-B.1.617.2) and four MeV plasmids (pMeV-WA1, pMeV-B.1.351, pMeV-B.1.1.7, and pMeV-B.1.617.2). In addition, the *preS-6P* gene of SARS-CoV-2 WA1, B.1.1.7, and B.1.617.2 was inserted into the MuV-JL1 strains which yielded three MuV-JL1 plasmids (pMuV-JL1-WA1, pMuV-JL2-B.1.1.7, and pMuV-JL2- B.1.617.2). All constructions were first identified by restriction enzyme digestion, PCR, and were confirmed by sequencing. Primers used for constructing these plasmids are listed in Table S3. The nucleotide sequences of *preS-6P* genes of SARS-CoV-2 VoCs are listed in Supplementary Data 1.

## Recovery of recombinant MuV and MeV
The recombinant MuV (rMuV-JL1 or rMuV-JL2) or MeV (rMeV) Edmonston strains expressing preS-6P of SARS-CoV-2 were recovered as described previously[33,57,58]. For MuV recovery, 2.5 μg of a plasmid encoding the full-length genome of MuV-JL2 or MuV-JL1 strain with the *preS-6P* gene and support plasmids (0.5 μg pN, 0.5 μg pP, and 0.5 μg pL) encoding the MuV genome-associated ribonucleocapsid complex were co-transfected into HEp-2 cells infected with a recombinant modified vaccinia Ankara virus (MVA-T7) expressing T7 RNA polymerase (kindly provided by Dr. Bernard Moss)[59]. Four-day later, the transfected cells were scraped off the plates and transferred together with the culture medium onto the 90% confluent Vero CCL81 cells for another 4 day of co-culturing to allow further amplification of the recovered recombinant virus. Subsequently, the recovered viruses were plaque purified as described previously. Individual plaques were isolated, and seed stocks were amplified in Vero CCL81 cells. Seed stocks were passed 2-3 times in Vero CCL81 cells and viral titers were determined by plaque assay performed in Vero CCL81 cells. The protocol for recovery of rMeV was identical to those described for rMuV with the exception of using a higher amount (5.0 μg) of the plasmid encoding the full-length genome of MeV with the *preS-6P* gene, and the support plasmids (1.5 μg pN, 1.5 μg pP, and 0.5 μg pL) encoding the MeV genome-associated ribonucleocapsid complex. All rMeVs were plaque purified and sequencing confirmed.

## MeV and MuV growth curves
Confluent Vero CCL81 cells were infected with individual rMuV or rMeV at a multiplicity of infection (MOI) of 0.1. After 1 h of adsorption, the inoculum was removed, the cells were washed twice with DMEM, fresh DMEM (supplemented with 2% FBS) was added, and the infected cells were incubated at 37 °C. At the indicated time points, cell lysates were subjected to freeze-thaw 3 times and combined with the cell culture fluid, and virus titers were determined by plaque assay in Vero CCL81 cells.

## MeV, MuV, and SARS-CoV-2 plaque assays
Confluent Vero CCL81 cells in 12-well plates were infected with 10-fold serial dilutions of rMuV or rMuV expressing SARS-CoV-2 S protein. After absorption for 1 h at 37 °C, cells were overlaid with 1 ml of DMEM containing 0.25% (w/v) low-melting temperature agarose, 0.12% (v/v) NaHCO$_3$, 2% (v/v) FBS, 25 mM HEPES, 2mM L-Glutamine, 100 μg/ml of streptomycin, and 100 U/ml penicillin. After incubation at 37 °C for 4 days (rMuV or rMeV) or 6 days (rMuV or rMeV expressing the preS-6P protein), cells were fixed with 4% paraformaldehyde for 2 h. The overlay was then removed, and the plaques were visualized by staining with 0.05% (v/v) crystal violet. SARS-CoV-2 plaque assay was performed on Vero-E6 cells in 12-well plate incubated for 2 days. The plaques were scanned using Image J Software.

## Detection of SARS-CoV-2 S protein by Western blot
Vero CCL81 cells were infected with rMuV-JL1, rMuV-JL2, or rMeV expressing preS-6P protein, as described above. At the indicated times post-infection, cells were lysed in RIPA buffer (Abcam, ab156034) on ice. Proteins were separated by 12% SDS-PAGE and transferred to a Hybond enhanced chemiluminescence nitrocellulose membrane (GE HealthCare, Chicago, IL, USA) in a Mini Trans-Blot electrophoretic transfer cell (Bio-Rad, Hercules, CA, USA). The blot was probed with rabbit anti-SARS-CoV-2 S polyclonal antibody (SinoBiological, Wayne, PA, USA, catalog 40150-T62-COV2) at a dilution of 1:2,000, followed by horseradish peroxidase (HRP)-labeled goat anti-rabbit secondary antibody at a dilution of 1:5,000. The blot was developed with SuperSignal West Pico chemiluminescent substrate (ThermoFisher Scientific), and developed and photographed by FluorChem Western blot imaging systems (Bio-Rad).

## Enzyme-linked immunosorbent assay (ELISA) for antibody measurement
SARS-CoV-2 S-specific binding antibodies in serum were assessed by ELISA described previously[30,32,34]. 96-well plates were coated with preS-6P (8 μg/ml) of SARS-CoV-2 WA1, B.1.617.2, B.1.1.7, B.1.351, or B.1.1.529 VoC in 50 mM Na$_2$CO$_3$ buffer (pH 9.6) and incubated at 4 °C overnight. After incubation, plates were washed once with wash buffer (0.05% Tween 20 in 1 × PBS) and blocked with 200 μl of 1% (w/v) Bovine Serum Albumin (BSA) per well at 4 °C overnight. After incubation, block solution was discarded, and plates were blotted dry. Serial dilutions of serum were added to wells, and plates were incubated for 1 h at room temperature before three more washes and 1 h incubation with horseradish peroxidase (HRP)-conjugated secondary antibody (goat anti-mouse IgG (H + L)) (1:15,000, Thermo Scientific, catalog no. 31430) or goat anti-hamster IgG (H + L) (1:15,000, Invitrogen, catalog no. PA1-28823). The IgA were detected by addition of HRP-conjugated anti-mouse IgA (Southern Biotech Associates Inc., Birmingham, AL) or HRP-conjugated anti-Hamster IgA (Brookwoodbiomedical, Jemison, AL). The Plates were washed three times, and 100 μl of SureBlue™ TMB 1-Component Microwell Peroxidase Substrate (Fisher Scientific, catalog no. 50-674-93) was added to each well; plate development was halted by adding 100 μl of H$_2$SO$_4$ (2 mol/L) per well. Endpoint titers were determined as the reciprocal of the highest dilution that had an OD$_{450}$ value 2.1-fold greater than the background level (normal control serum).

## Lentivirus pseudotyped neutralization assay
The SARS-CoV-2 pseudoviruses expressing a luciferase reporter gene were used to measure pseudovirus NAbs[39,60,61]. The HEK293T-hACE2 (human angiotensin-converting enzyme 2) cells ($2 \times 10^4$/well) were seeded in 96-well tissue culture plates overnight. The pseudoviruses harboring D614G, B.1.617.2, B.1.1.7, or B.1.351, Omicron BA.1, and

Omicron subvariant BA.4/5 S proteins were incubated with fourfold serial dilution of heat-inactivated serum at 37 °C for 1 h, followed by infection of HEK293T-ACE2 cells. The mutations in spike proteins of B.1.617.2, B.1.1.7, B.1.351, Omicron BA.1, and Omicron subvariant BA.4/5 were described in previous publications[60–62]. Gaussia luciferase activity in cell culture media was assayed 48 h and 72 h after infection. Note that, to ensure valid comparisons between SARS-CoV-2 variants, equivalent amounts of pseudovirus were used based on the pre-determined virus titers and samples of different variants were loaded side by side in each plate. SARS-CoV-2 NAb titer 50% ($NT_{50}$) for each sample was determined by non-linear regression with least squares fit in GraphPad Prism version 6.01.

### Recombinant antigen purification
The plasmids encoding soluble preS-6P protein (1-1273) of SARS-CoV-2 WA1, B.1.617.2, B.1.1.7, B.1.351, or B.1.1.529 were transfected into Free-Style™ 293-F cells to produce the preS-6P protein. The secreted preS-6P proteins in cell culture supernatants were purified via affinity chromatography. The purity of the protein was analyzed by SDS-PAGE and Coomassie blue staining. The protein concentration was measured using Bradford reagent (Sigma Chemical Co., St. Louis, MO, USA).

### Animals
Age-matched 6–8-week-old female and male specific-pathogen-free (SPF) interferon-alpha receptor 1 knockout (IFNAR1$^{-/-}$) mice were purchased from Jackson Laboratories (Bar Harbor, ME). 4–6 week-old SPF female golden Syrian hamsters were purchased from Envigo (Indianapolis, IN, USA). All animals were housed within ULAR facilities of The Ohio State University under approved Institutional Laboratory Animal Care and Use Committee (IACUC) guidelines (protocol no. 2009A1060-R3 and 2020A00000053). Each inoculation group was separately housed in rodent cages under animal biosafety level 2 (ABSL-2 for MeV and MuV) or ABSL3 (for SARS-CoV-2) conditions. All animals were housed at temperature of 22 °C with 53% humidity. Light dark cycle is 12 h light and 12 h dark.

### Animal Experiment 1: Comparison of the efficacy of the monovalent (rMuV-JL2-WA1) and trivalent vaccines in IFNAR1$^{-/-}$ mice
Twenty-five 4–6 week-old SPF female and male IFNAR1$^{-/-}$ mice were randomly divided into 5 groups (n = 5). Each group contains 3 female and 2 male mice. The mice from Group 1–4 were I.N. immunized with $1.2 \times 10^6$ PFU of trivalent vaccine TVC-I (Group1: rMuV-JL2-WA1, rMuV-JL2-B.1.1.7, and rMeV-B.1.351), trivalent vaccine TVC-II (Group2: rMuV-JL2-WA1, rMuV-JL2-B.1.1.7, and rMeV-WA1), monovalent vaccine (Group 3, rMu-JL2-WA1), or MMM vector control (Group 4, rMuV-JL1, rMuV-JL2 and rMeV). Group 5 was inoculated with same volume (30 µl) of DMEM. Three weeks later, all mice were boosted with the same virus or DMEM at the same dose, volume, and route. At weeks 2, 5, and 7 after immunization, blood samples were collected from each mouse by facial vein bleeding, and the serum was isolated for the detection of S-specific antibody by ELISA. At week 7, mice in Groups 1, 2, and 4 were transferred into BSL3 facility and challenged intranasally with $5 \times 10^4$ PFU of MA SARS-CoV-2 (MA10 strain). Mice in Group 5 continued to be housed in the BSL2 facility and were inoculated with 20 µL DMEM. After challenge, clinical signs and body weight of each mouse were monitored daily. At day 4 post-challenge, all mice were euthanized, the left lung was collected for detection of infectious SARS-CoV-2 by plaque assay.

### Animal Experiment 2: Comparison of the efficacy of the immunization routes in IFNAR1$^{-/-}$ mice
Forty 4–6 week-old SPF female and male IFNAR1$^{-/-}$ mice were randomly divided into 4 groups (Groups 1, 2, 4, n = 10, 5 females and 5 males; Group 3, n = 9, 5 females and 4 males). The trivalent vaccine TVC-III was composed of equal amounts ($4 \times 10^5$ PFU) of rMuV-JL2-WA1, rMuV-JL1-

B.1.617.2, and rMeV- B.1.351. Mice in Group 1 were immunized I.N. with $1.2 \times 10^6$ PFU of TVC-III. Mice in Group 2 were immunized S.C. with $1.2 \times 10^6$ PFU of TVC-III. Mice in Group 3 were immunized with $1.2 \times 10^6$ PFU of TVC-III via a combination of I.N. and S.C. route (half for I.N. and half for S.C.). Mice in Group 4 were immunized with $1.2 \times 10^6$ PFU of MMM vector control (rMuV-JL1, rMuV-JL2 and rMeV) via a combination of I.N. and S.C. route. Three week later, all mice were boosted with the same virus at same dose and route. At weeks 2, 5, and 7, blood samples were collected from each mouse by facial vein bleeding, serum isolated, and S-specific IgG antibodies were detected by ELISA. The sera at week 7 were used to detect SARS-CoV-2-specific neutralization antibody against lentivirus pseudotyped virus bearing spike of D614G mutation, B.1.617.2, B.1.351, Omicron BA.1, or Omicron subvariant BA. 4/5. At week 7, five mice from Groups 1, 2, and 4 were sacrificed, spleens from these three groups were collected for T cell assay. The remaining mice in Groups 1 (n = 5), 2 (n = 5), 3 (n = 4), and 4 (n = 5) were also sacrificed.

500 µl of sterile PBS was injected to the trachea, aspirate and inject the fluid back and forth 5 times, and the final bung bronchoalveolar lavage (BAL) were collected. After centrifugation at $3000 \times g$ for 5 min, supernatants were collected for detection of IgA and IgG titer by ELISA using preS-6P of SARS-CoV-2 WA1, B.1.617.2 or B.1.351 as the coating antigen.

### Animal Experiment 3: Comparison of the efficacy of the immunization doses in IFNAR1$^{-/-}$ mice
Twenty 4–6 week-old SPF female IFNAR1$^{-/-}$ mice were randomly divided into 4 groups (n = 5). The mice from Group 1–4 were I.N. immunized with a high dose ($1.2 \times 10^6$ PFU), a low dose ($3 \times 10^5$ PFU) of trivalent vaccine TVC-IV (rMeV-B.1.617.2, rMuV-JL1-B.1.1.7, and rMuV-JL2-WA1), a high dose ($1.2 \times 10^6$ PFU) of monovalent vaccine rMuV-JL2-WA1, or MMM vector control (rMuV-JL1, rMuV-JL2 and rMeV). Three weeks later, all mice were boosted with the same virus at the same dose, volume, and route. At weeks 2, 5, and 7 after immunization, blood samples were collected from each mouse by facial vein bleeding, and the serum was isolated for the detection of S-specific antibody by ELISA using preS-6P of SARS-CoV-2 WA1, B.1.1.7, B.1.351, or B.1.617.2 as the coating antigen. Week 7 sera were also used for the detection of IgA titer by ELISA using preS-6P of SARS-CoV-2 WA1 as the antigen. At week 7, anti-CD45-PE antibody was retroorbitally injected into mice 10 min prior to euthanasia to separate the tissue-resident (CD45-) and circulating (CD45 + ) T cells in the lungs. Mice then were euthanized for analysis of tissue-resident memory T cells in the lung.

### Animal experiment 4: Determine the immunogenicity of the trivalent vaccines in golden Syrian hamsters
Fifty 4 week-old female SPF golden Syrian hamsters were randomly divided into 4 groups. Hamster in Group 1 (n = 15) and Group 2 (n = 15) were inoculated with $1.2 \times 10^6$ PFU of trivalent vaccine TVC-V (rMuV-JL1-WA1, rMuV-JL2-B.1.1.7, and rMeV-WA1) and trivalent vaccine TVC-VI (rMuV-JL1-WA1, rMuV-JL2-B.1.617.2, and rMeV-WA1), respectively. Hamster in Group 3 (n = 15) and Group 4 (n = 5) were immunized with $1.2 \times 10^6$ PFU parental MMM vector, or the same volume of DMEM, respectively. The administration route was $6 \times 10^5$ PFU in 30 µL of DMEM for I.N. combined with $6 \times 10^5$ PFU in 500 µL of DMEM for S.C. Two immunizations were performed within a 3 weeks interval. At weeks 2, 5, and 7, blood samples were collected from each hamster via retro-orbital plexus, serum was isolated, and S-specific antibody against WA1, B.1.1.7, B.1.351, and B.1.617.2 was detected by ELISA. The serum at week 7 was isolated to detect neutralization antibody using a pseudotyped lentivirus bearing spike of SARS-CoV-2 WA1-D614G mutation, B.1.617.2, B.1.351, B.1.1.7, Omicron BA.1, and Omicron subvariant BA.4/5.

At week 4 after booster immunization, hamsters in Groups 1–3 were divided into 3 subgroups (n = 5) and challenged I.N. with $2 \times 10^4$

PFU of SARS-CoV-2 WA1, $2 \times 10^4$ PFU of B.1.617.2, or $7 \times 10^5$ PFU of Omicron BA.1. For Omicron BA.1 infection, hamsters were I.N. infected with $10^8$ PFU of Ad5-hACE2 5 days prior to the challenge with Omicron BA.1 virus. Following challenge, clinical signs and body weight of each hamster were monitored daily. Hamsters were sacrificed at day 4 (for challenge with SARS-CoV-2 WA1 or B.1.617.2) or day 3 (for challenge with Omicron BA.1). The left lung and nasal turbinate were collected from each hamster for detection of infectious SARS-CoV-2 by plaque assay. The right lung was preserved in 4% (v/v) phosphate-buffered formaldehyde for histology.

### Animal Experiment 5: Determine whether these co-viruses acted as adjuvants

Briefly, 4 week-old female hamsters were divided into 2 groups. Group 1 ($n = 5$) was immunized I.N. with $3 \times 10^5$ PFU of MuV-JL2-WA1 (diluted in DMEM) and Group 2 ($n = 5$) was immunized with a mixture of $3 \times 10^5$ PFU of MuV-JL2-WA1, null MeV, and null MuV JL1 viruses (each containing $10^5$ PFU). Three weeks later, each group was boosted with the same virus. At weeks 2, 5, and 7, serum was collected from each hamster to determine WA1-preS-6P specific IgG antibodies by ELISA.

### Animal Experiment 6: Determine the effects of doses and antigen compositions on immune response of trivalent vaccine

rMeV, rMuV-JL1, and rMuV-JL2 expresses the preS-6P of the original SARS-CoV-2 WA1 (rMuV JL2-WA1+rMuV-JL1-WA1 + rMeV-WA1) were combined to generate TVC-VII. rMeV, rMuV-JL1, and rMuV-JL2 expresses three different preS-6P proteins of three VoCs (rMeV-WA1 + rMuV-JL1-B.1.617.2 + rMuV-JL2-B.1.1.7) were combined to generate TVC-VIII. The immune responses of a low ($3 \times 10^4$ PFU), medium ($3 \times 10^5$ PFU), and high dose ($1.2 \times 10^6$ PFU) of TVC-VII and TVC-VIII in hamsters were compared. Briefly, 3 groups of 4 week-old female hamsters ($n = 5$) were immunized I.N. with TVC-VII at a low ($3 \times 10^4$ PFU), medium ($3 \times 10^5$ PFU), and high dose ($1.2 \times 10^6$ PFU) per hamster. Separately, another 3 groups of 4 week-old female hamsters ($n = 5$) were immunized I.N. with TVC-VIII at a low ($3 \times 10^4$ PFU), medium ($3 \times 10^5$ PFU), and high dose ($1.2 \times 10^6$ PFU) per hamster. Three weeks later, each group was boosted with the same virus at the same dose. At weeks 2, 5, and 7, sera were collected for the detection of IgG and IgA antibodies by ELISA using preS-6P of WA1, B.1.617.2, or B.1.1.7 as the coating antigen. In addition, week 7 sera were used for determination of NAb titer using pseudotyped virus neutralization assay.

### Animal Experiment 7: Determine the effects of preexisting MMM vector immunity on vaccine efficacy

Briefly, two groups of 4 week-old female hamsters ($n = 5$) were immunized S.C. with MMM vector (group 1, with pre-existing immunity) or DMEM (group 2, no pre-existing immunity) at week 0. At weeks 3 and 5, these two groups of hamsters were immunized I.N. with TVC VIII (rMeV-WA1 + rMuV-JL1-B.1.617.2 + rMuV-JL2-B.1.1.7). Serums were collected at weeks 2, 5, 7, 9, and 11 to examine the serum IgG and IgA antibody responses against WA1, variant B.167.2, and B.1.1.7 by ELISA. Sera were also used for determination of MuV and MeV-specific NAb by plaque reduction neutralization assay.

### Animal Experiment 8: Determine the effects of immune imprinting on immune response of the trivalent vaccine

Briefly, 20 4-week-old female hamsters were divided into 4 groups ($n = 5$). Groups 1, 2, and 3 received I.N. with two doses of rMuV-JL2-WA1 ($1.2 \times 10^6$ PFU) at weeks 0 and 3 to induce strong antibody against WA1. Group 4 received two doses of rMuV-JL2 vector control. For the third dose, Groups 1, 2, and 3 received I.N. with trivalent TVC-IX (rMeV-BA.1+rMuV-JL1-B1.617.2+rMuV-JL2-WA1), monovalent rMeV-BA1, and monovalent rMuV-JL2-WA1 at week 5, respectively. At weeks 2, 5, 7, and 9, sera were collected for the detection of IgG antibodies by ELISA using preS-6P of WA1, B.1.617.2, or BA.1 as the coating antigen. In addition, week 9 sera were used for determination of NAb titer using pseudotyped virus neutralization assay. At week 9, hamsters were intranasally infected with $10^8$ PFU of Ad5-hACE2 5 days, followed by challenge with $2 \times 10^5$ PFU of Omicron BA.1 virus. Following challenge, clinical signs and body weight of each hamster were monitored daily. Hamsters were sacrificed at day 3. The lung and nasal turbinate were collected from each hamster for detection of infectious SARS-CoV-2 titer by plaque assay.

### Flow cytometry analysis of antigen-specific cytokine producing T cells in spleen

CD4+ and CD8+ T cell responses were quantitated using SARS-CoV-2 S-specific peptide-stimulated intracellular cytokine staining assays[32,33,63]. A set of 181 peptides spanning the complete spike protein of the USA-WA1/2020 strain of SARS-CoV-2 (GenPept: QHO60594) were obtained from BEI Resources (catalog no. NR-52402). These peptides are 13–20 amino acids long, with 10 amino acid overlaps. For detection of SARS-CoV-2-specific intracellular cytokine production, $10^6$ cells were stimulated in 96-well round bottom plates with the S peptide pool (5 µg/ml), or media alone or PMA/Ionomycin (BioLegend, San Diego, CA, USA) as negative and positive controls, respectively, for 5 h in the presence of GolgiPlug (BD Biosciences, Franklin Lakes, NJ, USA). Following incubation, cells were surface stained for CD3, CD4, and CD8 for 30 min at 4 °C, fixed and permeabilized using the cytofix/cytoperm kit (BD Biosciences), and intracellularly stained for IFN-γ, TNF-α, IL-2, IL-17A, IL-21, IL-10, and IL-4 for 30 min at room temperature. The mouse reactive antibodies were from BioLegend, BD Biosciences, and ThermoFisher Scientific for analysis of T cells. Dead cells were removed using the LIVE/DEAD fixable Near-IR dead cell stain kit (Invitrogen). Events were collected on a BD LSRFortessa X-20 flow cytometer following compensation with UltraComp eBeads (Invitrogen). Data were analyzed using FlowJo v10 (Tree Star Inc., Ashland, OR, USA). Gating strategy is depicted in Fig. S17.

### Analysis of resident and circulating T cells in the lungs

Protocol for examining tissue-resident T cells was described previously[32,33]. To discriminate the resident and circulating T cells, anti-CD45-PE (Clone 30-F11, BD Biosciences) (3 µg in 100 µL sterile PBS) was retro-orbitally injected into mice 10 min prior to euthanasia to label circulating lymphocytes, while resident lymphocytes are protected from labeling (7, 8). Peripheral blood was collected at time of sacrifice and checked by flow cytometry to confirm that >90% of circulating lymphocytes were CD45-PE+. Lungs were isolated and processed into a single cell suspension using the gentleMACS tissue dissociator and mouse lung dissociation kit (Miltenyi Biotec, Auburn CA, USA). The single cells were resuspended in T cell media (RPMI 1640 supplemented with 0.1% gentamicin antibiotic, 10% HI-FBS, Glutamax, and $5 \times 10^{-5}$ M β-ME) and incubated for 4−5 h at 37 °C with protein transport inhibitor cocktail (eBioscience, San Diego, CA, USA). Cells were either stimulated nonspecifically with PMA (50 ng/ml) /Ionomycin (500 ng/ml) or with two Spike peptide pools covering the C- and N- terminus (PepTivator SARS-CoV-2 Prot_S1 and PepTivator SARS-CoV-2 Prot_S + ) (Miltenyi Biotec catalog no. 130-126-701& 130-126-700), Peptide pools were used at a final concentration of 1 µg/ml each peptide. Cells incubated with DMSO alone were used as negative control. Following stimulation, cells were washed with cold PBS prior to staining with Live/Dead Zombie NIR fixable viability dye (BioLegend, catalog no. 423105) for 30 min at 4 °C. Cells were then washed twice with PBS supplemented with 1% heat inactivated FBS (1% FBS) (FACS buffer) and resuspended in Fc Block (clone 93) (eBioscience, catalog no. 14-0161-86) at 4 °C for 5 min before surface staining with a mixture of the following Abs for 20 min at 4 °C: CD3 V450 (clone 17A2, 1:500 dilution, BD Biosciences, catalog no. 561389), CD4 BV750 (clone H129.19, 1:1,000 dilution, BD Biosciences, catalog no. 747275), CD44 PerCP-Cy5.5 (clone IM7, 1:500 dilution, BD Biosciences, catalog no. 560570),

CD62L BV605 (clone MEL-14, 1:2,000 dilution, BD Biosciences, catalog no. 563252), CD69 BV711 (clone HI.2F3, 1:500 dilution, BD Biosciences, catalog no. 740664). After two washes in FACS buffer, cells were resuspended in intracellular fixation buffer (eBioscience, catalog no. 00-8222-49) and incubated for 20 min at room temperature (RT). Following permeabilization (eBioscience, catalog no. 00-8333-56), intracellular staining (30 min at 4 °C) was done using a mixture of the following Abs: IFN-g FITC (clone XMG1.2, 1:125 dilution, eBioscience, catalog no. 11-7311-82), IL-17 PE-Cy7 (clone eBio17B7, 1:125 dilution, eBioscience, catalog no. 25-7177-82), and IL-5 APC (clone TRFK5, 1:125 dilution, BD Biosciences, catalog no. 554396). To identify CD8$^+$ T cells, the same panel was used with CD8 APC (clone 53-6.7, 1:1,000 dilution, BioLegend, catalog no. 100712) and IFN-g FITC only. Fluorescence minus one or isotype control antibodies were used as negative controls. Samples were collected on a Cytek Aurora flow cytometer (Cytekbio, Fremont, CA, USA). Analysis was performed using FlowJo software, version 10.8.0. The number of cells within each population was calculated by multiplying the frequency of live singlets in the population of interest by the total number of cells in each sample. Gating strategy for CD4$^+$ and CD8$^+$ T cells stimulated by S peptide is depicted in Fig. S18 and Fig. S19, respectively. Gating strategy for CD4$^+$ and CD8$^+$ T cells stimulated by PMA/Ionomycin is depicted in Fig. S20 and Fig. S20, respectively.

### SARS-CoV-2 titration in animal tissues
After challenge, animals were euthanized. The lung of each mouse and the left lung lobe and nasal turbinate were weighed and homogenized in 1 mL of sterile PBS. The SARS-CoV-2 viral titer in these tissues was determined by plaque assay in Vero-E6 cells[32,33].

### Histological analysis of lung tissues
The right lung lobe from each hamster was fixed in phosphate-buffered 4% (v/v) formaldehyde for 14 days and then transferred out of the BSL-3 facility. Tissues were then embedded in paraffin, sectioned at 5 μm in duplicate, deparaffinized, and rehydrated. Each section was stained with hematoxylin-eosin (H.E.) and evaluated a pathologist for blind review of histological changes. Criteria for pathology scoring system was described previously[30–32,34].

### Statistical analysis
Statistical analysis was performed by two-sided Student $t$-test, one-way, or two-way ANOVA multiple comparisons using GraphPad Prism version 6.01. A $P$-value of <0.05 was considered statistically significant.

### Reporting summary
Further information on research design is available in the Nature Portfolio Reporting Summary linked to this article.

## Data availability
The experimental data generated in this study are provided in the main text, Figures, or in the Supplementary Information/Source Data File. Source data are provided with this paper.

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

## Acknowledgements

This study was supported by startup fund and bridge funding (J.L.) from the Department of Veterinary Biosciences, College of Veterinary Medi-cine at The Ohio State University and a seed grant (M.E.P. and J.L.) from the Abigail Wexner Research Institute at Nationwide Children's Hospital. M.C. was supported by an NIH T32 training grant (T32AI165391). J.X. and Y.Z. were supported by C. Glenn Barber Fund Trust. This study was also supported in part by grants from the NIH (R01AI090060 and P01AI175399 to J.L.; P01 AI112524 to M.E.P. and J.L; and R01AI093848 and U19AI42733 to M.E.P.). P.N.B. was supported by NIH grants R01AI145144 and R01AI157205. MMS was supported by a scho-larship from the Egyptian Ministry of Education. S.L.L was supported by an anonymous private donor to OSU and by the National Cancer Institute U54CA260582. The content is solely the responsibility of the authors and does not necessarily represent the official views of the National Institutes of Health. We thank Jason McLellan (University of Texas at Austin) for providing plasmids for expressing stabilized prefusion spike proteins of SARS-CoV-2 WA1 and Delta variant for this study. We thank the BSL3 working group and University Laboratory Animal Resources staff at The Ohio State University for their support for this study. We thank Sally L. Li, a high school student at Columbus Academy, for drawing cartoon images of a mouse and hamster for the figures. We

thank members of the J.L. laboratory for technical help and critical readings of the manuscript.

## Author contributions

J.L., Y.Z., M.C., J.X., P.N.B., S.L.L, and P.D. designed research; Y.Z., J.X., M.C., P.Q., M.M.S., S. J.Y., J.M., I.T., M.KC., J.M.H., Y.A.G., J.P.E., M.L., C.Y., C. C.H., X. L., L.M.-S., P.N.B., P.D. and J.L. performed research; L.M.-S., J.S.Y., M.E.P., S.L.L, P.D. and J.L. contributed new reagents/analytic tools; Y. Z., M.C., J.X., P.Q., S.J.Y., M.M.S., P.N.B., M.E.P., S. L.L, P.D. and J.L. analyzed data; J.L. Y.Z., M.C., J.X. and wrote the paper; all other authors edited the manuscript.

## Competing interests

The Ohio State University has filed an invention report on this approach to the development of intranasal trivalent SARS-CoV-2 vaccines. J.L., Y.Z., J.X., M.C. and X.L. are the inventors. Other authors do not have competing interests.
