## [Peer Review File · Nature Communications]

REVIEWER COMMENTS

Reviewer #1 (Remarks to the Author):

Review of paper by Zhang, Li et al,

The paper by Zhang, Li, and colleagues Three SARS-CoV-2 spike protein variants 1 delivered intranasally by measles and 2 mumps vaccines are broadly and highly protective was an intriguing approach, but complicated given the number of permutations and combinations that could be tested. I couldn't easily follow the results and the figures without the group keys right next to each other. I know it's impractical to test all possible combinations, but because of this, the experiments came across more like a here's "a one off result with this combination", without showing anything interesting about testing hypothesis such as co-administration of a variant using one vector with a parental vaccine with a different vector. My enthusiasm was high for the paper before reading the manuscript, but was much less after reading the paper. Here's three major ways to improve the paper:

1. To ask fundamental questions of biology, I would have reduced the doses and determined if addition of a variant with the original parental led to improved breadth. I would have also tested the original parental with null co-viruses to determine if the co-viruses were acting as an adjuvant, or actually providing additional antigenic stimulation. At the excessive doses tested, the antibody responses simply just overwhelmed the system to allow any interpretation of the necessity of a variant. A better experiment would also compare 3 viruses expressing the parental with 3 viruses all expressing a different variant to demonstrate necessity of variants. Again, lower doses would be needed to keep from overwhelming the system.
2. If the goal was to develop a novel vaccine modality, significantly more work would be needed to justify using not just one GMP product, but potentially 3-6. That's a big ask and would be impractical unless there's a fundamental improvement beyond what else is out there. As stated in #1, I would really make sure all those different products were necessary. One key experiment would be using higher doses of 1 product to see if it could compensate make a similar response to lower doses of 3 products. One issue is that the authors seem to be using higher doses in a mouse than typically used in humans, so some justification would be needed.
3. A critical step for any new modality, is to test against existing approved approaches. It wasn't possible to obtain mRNA vaccines before, but it can be done now. Comparing to mRNA vaccines for immune responses might be critical to advance this platform. If it isn't possible to test against mRNA because of the mouse model, so state in the discussion.

More detailed comments:

Results

1. Figures 1-3 are not needed in the main body of the paper. The paper should focus on the immunogenicity and protection from viral challenge. The early figures should be used in a different paper on the making the vaccine candidates or put in the supplemental.
2. While the use of the trivalent seems to have improved the results of using the monovalent vaccine, what would have been better would be to compare to null 2nd or 3rd viruses. Is there an adjuvant effect?

3. Figure 4E and F. Is it possible for key?
4. Figure 5. What the signal at day 0? Is it always 2 log₁₀? The mucosal responses seem to look the same as the IgA Serum responses in figure 4F. Is this due to blood contamination in the preps?
5. Figure 6 . The IFN γ responses seem low relative to TNF α . If this is the real response, then perhaps a comparison between other COVID-19 mouse studies would be appropriate in the discussion.

Discussion.

6. There needed to be a discussion of the animal models and vaccine doses, and relevance to humans. Mice are substantially smaller animals and using full human doses is already substantial overkill. In fact, the authors used much higher doses in mice than humans. The authors were using 1e6 PFU, and humans get 2-3 logs lower. Further, the use of IFN- α KO mice to promote replication seems likely to cause interpretation problems when moving to humans. How relevant is the mouse model to humans?
7. There was no discussion about how the hamster results compared to other hamster studies with mucosal vaccines. Are these results similar, better? Bricker, et al, Cell Reports; Braun, et al, Frontiers in Immunology, are some examples.
8. Omicron is weaker virus in hamsters than the other challenge viruses. A brief mention of this seems important in interpreting any results on the virus. The authors were focused on the fact that their vaccine protected in the absence of nAbs, but this might have been simply because of the weaker challenge.
9. Lots of discussion of the benefits of the replicating measles platform, but there was no mention of the fact that Merck dropped their SARS-COV-2 vaccine approach because it didn't work. That might be important for a balanced discussion.
10. There really needs to be a paragraph on the limitations of the results. The studies were complicated and doses were relatively high for the size of the animal.

Methods

1. The method described for obtaining the IgA in the lungs is likely to create contamination from the blood, so it cannot be certain that the response is mucosal or simply monomeric IgA from the blood. Typically, the approach is to flush the lungs carefully at harvest with PBS or to flush the circulatory system before harvesting lungs and grinding them up.

From the package insert of the MMR vaccine.

After reconstitution, each approximately 0.5 mL dose contains not less than 3.0 log₁₀ TCID₅₀ (tissue culture infectious doses) of measles virus; 4.1 log₁₀ TCID₅₀ of mumps virus; and 3.0 log₁₀ TCID₅₀ of rubella virus.

Reviewer #2 (Remarks to the Author):

The authors describe the generation and preclinical use of MMR-based measles and mumps vectored S6P-expressing live-attenuated COVID-19 vaccine candidates. Measles and mumps vectors that express S6Ps derived from ancestral, alpha, beta, and delta variants were generated. The resulting viruses could be propagated to high titers although they were attenuated compared to the parental viruses. Intranasal immunization with trivalent mixtures (JL1 + JL2 + Me) was protective in IFNAR^{-/-} mice against challenge with mouse-adapted SARS-CoV2. Spike-specific IgG, IgA, T cell responses, and Trms were generated by intranasal immunization of IFNAR^{-/-} mice with TVC-III. TVC combinations

also protected hamsters against challenge with WA1, Delta, and BA.1 with close to background viral replication observed in the nose after WA1 and Delta challenge.

The manuscript is well written and covers an extensive set of experiments.

Major comments:

1. In figure 3 and 7, a control group immunized with MeV-WA1 + JL1-WA1 + JL2-WA1 is required to support the possible benefit of a TVC vaccine that comprises two or three different S6Ps over a mono-specific vaccine in this model.
2. In Fig. 9, a control group immunized with MeV-WA1 + JL1-WA1 + JL2-WA1 is required to support the possible benefit of the TVC-V and -VI over the monospecific MMM-vectored S2P vaccine.
3. Most individuals have been vaccinated with MMR in childhood and thus likely have pre-existing immunity against the MMM-vectored Spike vaccine candidates presented in this work. Spike-specific immune responses induced by i.n. immunization with a TVC combination after priming of mice or hamsters by s.c. immunization with MMM should be compared with such responses in non-primed animals.
4. Line 1088: "... or the same volume of DMEM". Please clarify where the results of this DMEM control immunized group are presented in the Results section.

Minor comments;

1. Line 191: S6P ... secreted Please clarify whether the S6P transgenes were designed for secretion (lacked the C-terminal transmembrane domain?).
2. What does "normal" mean in Fig.4 G and Fig.10 A, E, and I?
3. Line 384: "IFN-g-producing" repeated.
4. Line 626: multivalent vaccines to prevent influenza are mentioned as a successful approach. It is, however, generally accepted that this success is limited with vaccine effectiveness of 20-60% and as low as 5 to 39% against the H3N2 component in influenza vaccines (McLean and Belongia 2021). Please provide another example of multivalent vaccines that are relatively successful. Perhaps HPV vaccines?
5. Line 823: hamsters in each group in Fig.9A instead?

Reviewer #3 (Remarks to the Author):

In this manuscript, Zhang et al design and test trivalent vaccines consisting of measles or mumps viruses harboring different SARS-CoV-2 spikes. These vaccines are an extension of a series of publications from the authors wherein they have tested multiple SARS-CoV-2 vaccine modalities. They demonstrate that these trivalent vaccines result in greater antibody and T-cell responses than monovalent, and importantly, elicits mucosal responses which are poorly elicited by the currently utilized mRNA vaccines, and that this corresponds to protection from viral challenge. These effects were shown in both IFNAR^{-/-} mice and in golden Syrian hamsters. While the presented designs were only able to generate weak responses against Omicron, the authors note that this platform is easily modifiable and can be updated with the latest variants. Therefore, these vaccines may serve as a component of future COVID-19 vaccination efforts.

General comment:

The proposed approach is unique and interesting and the authors should be commended for conducting a wealth of in vivo experiments which show the robust effects of their vaccines. As the COVID-19 pandemic continues, next-generation vaccines are surely needed, and the authors are correct in noting that the current vaccines suffer from their inability to elicit strong mucosal responses, which are important for protection. The manuscript is well-written and clear.

However, the setting that the authors have tested their trivalent vaccines in this manuscript is unfortunately perhaps not the most relevant or clinically significant. At this point in the pandemic, virtually the entire world has encountered SARS-CoV-2 multiple times, either through vaccination and/or infection. This has of course, resulted in numerous variants arising due to the immune pressure, but has also resulted in strong immune imprinting, as has now been shown by many groups. Recent reports suggests that as a result of this immune imprinting, the currently utilized bivalent mRNA vaccines have dampened effects against new variants and it has been suggested that instead, monovalent vaccines without WA1 designed against the most recent variants would be better. Thus, the experiment that the authors should conduct to demonstrate that their trivalent strategy is effective and does not suffer from these imprinting problems that the bivalent mRNA vaccines face is to provide a trivalent vaccination (with WA1 and variants) as a booster to animals that have already received two or three doses of a monovalent WA1 vaccine and compare this to providing a monovalent Omicron booster in terms of breadth and potency of response.

Major comments:

- It would be helpful to the scientific community to provide the sequences of the vaccines as a supplement or deposited to GenBank.
- It is noted that there are 48 possible trivalent vaccine combinations in lines 254-254, but there is no explanation provided for why the six chosen combinations were chosen. To this reviewer it is not very clear why these specific combinations were chosen. Perhaps the authors can add their rationale for each of the combinations.
- There is a focus on the current generation of vaccines not providing robust mucosal immunity and consequently the described vaccines were developed, but it would be helpful if such a group (e.g., vaccinated with BNT162b2 or mRNA-1273) was included as a comparator to know how much greater the mucosal immunity generated by these vaccines is.
- Figures 2C and 3C: Is the antibody that is being used in these blots specific for S or does it also bind S1 and S2? Are the other bands shown here S1/S2? If so, why do the three vectors have differing cleavage ratios?
- In the experiments shown in Figure 4, 7, and 9, it seems unexpected that the monovalent vaccine, which has 1.5 or 3 times the amount of this antigen than the trivalent vaccines, still has a lower autologous response. E.g., in Figure 4, why does rMuV-JL2-WA1 alone not have a higher WA1-directed response than the two trivalent groups? Perhaps the authors can add an explanation for this to the Discussion.
- Figures 4, 5, 7, 9: It is confusing and makes it difficult to compare to have IgG in log10 but IgA in

log₂ units, please place them on the same scales.

- Figure 7: A vector control group is mentioned in the text but not shown in any of the panels. Why is there seemingly no difference between the low and high dose of vaccine?
- Figure 8: Why does the monovalent group have less CD4+ Trms than the trivalent group? In particular, why would it have less than the low dose trivalent group?

Minor comments:

- Figures 1D, 2D, 3D, S1, S2, S3: Please add a scale bar.
- Figure 4G: What is the blue "ns"?
- Figure 4: This reviewer suggests to maintain consistency in coloring throughout the panels; e.g., the green in A through F is different than the green in G and H
- Figure 5D, 9H: It may be apt to label Alpha, Beta, and Delta as B.1.1.7, B.1.351, and B.1.617.2 given that they are labeled as such in all other panels.
- Figure 5E, 5F, 5G: The y-axes should be kept the same to allow for easier comparisons between panels.
- Figure 10K: The y-axis should be kept the same as other panels to allow for easier comparisons between panels.
- Figure 11: This figure could perhaps be moved to the supplement.
- Lines 893-900: Please provide the accession numbers for the sequences of the viruses used.
- Lines 992-1012: The mutations that were introduced to produce the variant pseudoviruses and spikes should be noted (as not everyone has used the same mutations for the variants, particularly Omicron).
- Line 1178: A reference seems to be missing here.
- References 38 and 60 appear to be the same.

Responses to the reviewers' comments

Introduction: We thank the three reviewers for their careful review of our manuscript. Their comments are extremely helpful in improving this manuscript. In an attempt to address the reviewers' major comments, we have conducted 4 new Animal Experiments. These animal experiments took quite a long time, which delayed the resubmission of the manuscript. Here are the brief answers to four major concerns raised by the reviewers:

1. Efficacy of the lower doses (Reviewer #1). We have now compared a low (3×10^4 PFU), medium (3×10^5 PFU), and high dose (1.2×10^6 PFU) of two trivalent vaccines side by side. A low dose of 3×10^4 PFU is sufficient to induce strong serum IgG, IgA, and neutralizing antibody (NAb). Please see new Fig.8, Results section lines 532-550, Discussion section lines 773-791.

2. Necessity (benefit) of including 2 or 3 spikes for inducing broad NABs (Reviewers#1 and 2). In our new hamster studies, we directly compared WA1-, B.1.617.2-, and B.1.1.7-specific IgG, IgA, and NAb titer between 3 viruses expressing 3 different preS-6P proteins (rMeV-WA1 + rMuV-JL1-B.1.617.2 + rMuV-JL2-B.1.1.7) and 3 viruses expressing one same preS-6P (rMuV-JL2-WA1 + rMuV-JL1-WA1 + rMeV-WA1). We found no significant difference between them. Because the antigenic distance among WA1, B.1.617.2, and B.1.1.7 is small, it may not be necessary to include B.1.617.2 or B.1.1.7 spike in the vaccine. In a separate study, we recently showed that another trivalent vaccine (rMeV-BA.1, rMuV-JL1-B.1.617.2, and rMuV-JL2-WA 1) induced broad NABs against Omicron BA.1, BA.4/5, B.1.617.2, and D641G whereas monovalent vaccine (rMeV-BA.1) only neutralized BA.1 but not BA.4/5 or B.1.617.2 variant in both IFNAR-/- mice and hamsters (Xu et al., PNAS, 2023). Because the antigenicity of Omicron BA.1 is distinct from the original WA1 and B.1.617.2, it is necessary to include the spike of BA.1 to induce broad neutralizing against D641G, B.1.617.2, and BA.1. Thus, the necessity of including 2 or 3 spikes of variants depends on the antigenic distance of these variants. We have included new data (new Fig.8, Fig.S12, 13, 14, and 15), Results section lines 552-566, and Discussion section lines 802-818).

3. The impact of preexisting MMM vector immunity on SARS-CoV-2-specific antibody induced by the trivalent vaccine (Reviewer #2). Our new animal studies showed that the preexisting MMM vector immunity had a minimal impact on SARS-CoV-2-specific antibodies induced by trivalent vaccines. We have included new data (new Fig.9), Results section lines 567-588, and Discussion section lines 676-680.

4. Immune imprinting issue of the trivalent vaccine (Reviewer #3). In our new hamster studies, 3 groups of hamsters were first received two doses of rMuV-JL2-WA1 to induce strong WA1-specific antibody, and followed by a third dose of a trivalent vaccine (Group 1, containing WA1, B.1.617.2, and BA.1), a monovalent BA.1 (Group 2), or a monovalent WA1 (Group 3). We found that Group 2 induced a 2.5-fold higher BA.1-specific NAb than Group 1 ($P < 0.05$). Therefore, the BA.1-specific antibody response of the trivalent vaccine may be affected by the immune imprinting induced by the previous WA1 preS-6P immunization. However, the lower BA.1-specific IgG induced in the trivalent vaccine may be caused by the dose effect because the dose of rMeV-BA1 in trivalent vaccine (Group 1) is 3 times less than that in monovalent rMeV-BA1 (Group 2). We have included these data and discussed these results. We have now included these new data (new Fig.10, Fig.S16, Results section lines 589-629) and Discussion section lines 819-834.

In summary, the manuscript proved the concept of using a trivalent vaccine carrying 3 spikes of 3 different SARS-CoV-2 variants. This strategy is particularly useful for preventing variants that are antigenically distinct from the ancestral SARS-CoV-2 WA1 strain and previous variants. This platform is easily modifiable and can be updated with the latest variants. For example, using 3 viruses expressing preS-6P of HV.1, EG.5, and XBB.1.5 to prevent the current co-circulating variants.

More detailed point-to-point responses are summarized below:

Reviewer 1

1. To ask fundamental questions of biology, I would have reduced the doses and determined if addition of a variant with the original parental led to improved breadth (**Point 1a**). I would have also tested the original parental with null co-viruses to determine if the co-viruses were acting as an adjuvant, or actually providing additional antigenic stimulation (**Point 1b**). At the excessive doses tested, the antibody responses simply just overwhelmed the system to allow any interpretation of the necessity of a variant (**Point 1a**). A better experiment would also compare 3 viruses expressing the parental with 3 viruses all expressing a different variant to demonstrate necessity of variants (**Point 1c**). Again, lower doses would be needed to keep from overwhelming the system (**Point 1a**).

Response: **Regarding Point 1b** (whether these co-viruses acted as adjuvants): We have conducted a new hamster experiment to address this comment. Briefly, one group (n=5) was immunized with only rMuV-JL2-WA1 with DMEM and another group (n=5) was immunized with MuV-JL2-WA1 and null MeV and MuV JL1 viruses. Three weeks later, each group was boosted with the same virus. After analyzing the sera collected from these groups at weeks 2, 5, and 7 using ELISA to measure WA1-preS-6P specific IgG antibodies, we found that while the rMuV-JL2-WA1 + null MeV + null MuV JL1 group had a significantly lower than the rMuV-JL2-WA1 at week 2 ($P < 0.0001$), there was no significant difference between the two groups at weeks 5 and 7 ($P > 0.05$) (new Fig.S11). Thus, the addition of these co-viruses does not act as an adjuvant. Instead, it interferes with SARS-CoV-2-specific antibody response at the early time point. These new results were reported in new Supplementary Fig.S11, and Results section lines 520-530.

Fig.S11. The effect of the parental MeV and MuV on immune responses of rMuV-JL2-WA1. Two groups ($n=5$) of 4-week-old female hamsters were immunized with 3×10^5 PFU of MuV-JL2-WA1 (Group 1) or a mixture of 3×10^5 PFU of MuV-JL2-WA1, null MeV, and null MuV JL1 viruses (each containing 10^5 PFU) (Group 2). Three weeks later, each group was boosted with the same virus. At weeks 2, 5, and 7, serum was collected from each hamster to determine WA1-preS-6P specific IgG antibodies by ELISA. Student's t -test was used for statistical analysis ($ns > 0.05$; $****P < 0.0001$).

Regarding Point 1a and c (whether a low dose can trigger an effective immune response; whether compare 3 viruses expressing the parental with 3 viruses all expressing a different variant to demonstrate the necessity of variants). This is the same as Point 2 (see our response below).

2. If the goal was to develop a novel vaccine modality, significantly more work would be needed to justify using not just one GMP product, but potentially 3-6. That's a big ask and would be impractical unless there's a fundamental improvement beyond what else is out there. As stated in #1, I would really make sure all those different products were necessary. One key experiment would be using higher doses of 1 product to see if it could compensate make a similar response to lower doses of 3 products. One issue is that the authors seem to be using higher doses in a mouse than typically used in humans, so some justification would be needed. **This is the same as Point 1a and c**

Response: We have now conducted new hamster immunization experiments to address **Point 1a and c and Point 2**. These comments can be summarized as two questions (1) the effectiveness of lower doses, and (2) the necessity of including spikes of variants in trivalent vaccine design.

In the new experiment, we used a trivalent vaccine candidate (TVC-VII): 3 viruses expressing the original SARS-CoV-2 WA1 preS-6P (rMuV-JL2-WA1 + rMuV-JL1-WA1 + rMeV-WA1); and TVC-VIII: 3 viruses expressing three different preS-6P proteins of three variants of concern (rMeV-WA1 + rMuV-JL1-B.1.617.2+ rMuV-JL2-B.1.1.7). Also, as suggested by the reviewer, we compared the efficacy of a low (3×10^4 PFU), medium (3×10^5 PFU), and high dose (1.2×10^6 PFU) per hamster for TVC-VII and TVC-VIII.

Experimental design: 3 groups of hamsters ($n=5$) were immunized TVC-VII at a low (3×10^4 PFU), medium (3×10^5 PFU), and high dose (1.2×10^6 PFU) per hamster. Separately, 3 groups of hamsters ($n=5$) were immunized TVC-VIII at a low (3×10^4 PFU), medium (3×10^5 PFU), and high dose (1.2×10^6 PFU) per hamster. Three weeks later, each group was boosted with the same virus at the same dose. At weeks 2, 5, and 7, sera were collected for the detection of IgG and IgA antibodies and neutralizing antibodies (NAb). These results are shown in **new Fig.8.** and **Results section lines 532-550.**

(1) Regarding the dose effects: We analyzed the efficacy of three different doses. We found that all three doses (low, medium, and high) induced strong serum IgG, serum IgA, and NAb. For serum IgG and serum IgA, no significant difference was observed between medium and high doses at three time points (weeks 2, 5, and 7) using three preS-6P of WA1, B.1.617.2, and B.1.1.7. However, the IgG and IgA titer in the low dose was significantly lower than the high dose at most time points using three preS-6P of WA1, B.1.617.2, and B.1.1.7. Week 7 sera from low and high doses were chosen to examine NAb. Importantly, although the high dose had a higher NAb than the low dose for both TVC-VII and TVC-VIII, there is no significant difference between low and high doses against all variants (D614G, B.1.617.2, B.1.1.7,

Omicron BA.1, or BA.4/5). **These results demonstrate that a low dose of 3×10^4 PFU was sufficient to induce strong SARS-CoV-2-specific NABs.**

Fig.8. The effects of doses and antigen compositions on the immune responses of trivalent vaccines. 30 female hamsters were randomly divided into 6 groups ($n=5$). The first 3 groups (A-F) were immunized *i.n.* with 3×10^4 , 3×10^5 , and 1.2×10^6 PFU of TVC-VII (rMeV-WA1+rMuV-JL1-B.1.617.2+rMuV-JL2-B.1.1.7), and the other 3 groups (G-L) were immunized *i.n.* with 3×10^4 , 3×10^5 , and 1.2×10^6 PFU of TVC-VII (rMuV-JL2-WA1+rMuV-JL1-WA1+rMeV-WA1). Three weeks later, each group was boosted *i.n.* with the same dose of the same vaccine. WA1 (A, G), B.1.617.2 (B, H), and B.1.1.7 (C, I) specific serum IgG titers were determined by ELISA. Week 5 (D, J) and week 7 (E, K) serum IgA titers were determined by ELISA. Week 7 sera were used for the determination of NABs (F, L) against different VoCs by pseudotyped virus neutralization assay.

(2) Regarding the necessity of including spikes of variants in the trivalent vaccine design: The above data (Fig.8) were re-analyzed by comparing the WA1-, B.1.617.2-, and B.1.1.7-specific serum IgG, IgA, and NAb titers between TVC-VII and TVC-VIII. This analysis showed no significant difference in WA1-, B.1.617.2-, and B.1.1.7-specific serum IgG or IgA titer between TVC-VII and TVC-VIII at all three doses (Fig.S12, 13, and 14, respectively). Similarly, there is no significant difference in WA1-, B.1.617.2-, or B.1.1.7-specific NAb titer between TVC-VII and TVC-VIII at low (Fig.S15A) and high (Fig.S15B) doses. Because the antigenicity of B.1.617.2, B.1.1.7, and WA1 are close, preS-6P of WA1 alone is sufficient to

neutralize B.1.617.2 and B.1.1.7. Thus, when antigenicity of variants is close, it may not be necessary to include multiple spikes in trivalent vaccine design. Please see Results section lines 552-566.

However, it should be noted that TVC-VII (3 viruses expressing WA1) is different from a monovalent virus expressing WA1 (for example rMuV-JL2-WA1 alone). Although the primary target of both MeV and MuV is the respiratory tract, they have different host tropisms and may infect different tissues, organs, and cell types when they are delivered intranasally. It is possible that MeV, MuV-JL1, and MuV-JL2 carrying WA1 had synergetic effects thereby inducing a strong immune response. We have discussed this point. Please see Discussion section lines 660-668.

- rMuV-JL1-WA1+rMuV-JL2-WA1+rMeV-WA1(3×10^4 PFU)
- rMuV-JL1-B.1.617.2+rMuV-JL2-B.1.1.7+rMeV-WA1 (3×10^4 PFU)

Fig.S12. Comparison of WA1-, B.1.617.2-, and B.1.1.7-specific IgG and IgA titers of TVC-VII and TVC-VIII at an immunization dose of 3×10^4 PFU. The serum IgG data at a dose of 3×10^4 PFU in Fig.8 were re-analyzed by comparison of WA1 (A)-, B.1.617.2 (B)-, and B.1.1.7 (C)-specific IgG titer. Similarly, serum IgA data at a dose of 3×10^4 PFU in Fig.8 were re-analyzed by comparison of WA1 (D)-, B.1.617.2 (E)-, and B.1.1.7 (F)-specific IgG titer. Student's *t*-test was used for statistical analysis (ns > 0.05; **P* < 0.05; ***P* < 0.01).

- rMuV-JL1-WA1+rMuV-JL2-WA1+rMeV-WA1(3×10^5 PFU)
- rMuV-JL1-B.1.617.2+rMuV-JL2-B.1.1.7+rMeV-WA1(3×10^5 PFU)

Fig.S13. Comparison of WA1-, B.1.617.2-, and B.1.1.7-specific IgG and IgA titers of TVC-VII and TVC-VIII at an immunization dose of 3×10^5 PFU. The serum IgG data at a dose of 3×10^5 PFU in Fig.8 were re-analyzed by comparison of WA1 (A)-, B.1.617.2 (B)-, and B.1.1.7 (C)-specific IgG titer. Similarly, serum IgA data at a dose of 3×10^5 PFU in Fig.8 were re-analyzed by comparison of WA1 (D)-, B.1.617.2 (E)-, and B.1.1.7 (F)-specific IgG titer. No significant difference in VoC-specific IgG or IgA titer is observed between TVC-VII and TVC-VIII. Student's *t*-test was used for statistical analysis (ns > 0.05; ***P* < 0.0001).

- rMuV-JL1-WA1+rMuV-JL2-WA1+rMeV-WA1 (1.2×10^6 PFU)
- rMuV-JL1-B.1.617.2+rMuV-JL2-B.1.1.7+rMeV-WA1 (1.2×10^6 PFU)

Fig.S14. Comparison of WA1-, B.1.617.2-, and B.1.1.7-specific IgG and IgA titers of TVC-VII and TVC-VIII at an immunization dose of 1.2×10^6 PFU. The serum IgG data at a dose of 1.2×10^6 PFU in Fig.8 were re-analyzed by comparison of WA1 (A)-, B.1.617.2 (B)-, and B.1.1.7 (C)-specific IgG titer. Similarly, serum IgA data at a dose of 1.2×10^6 PFU in Fig.8 were re-analyzed by comparison of WA1 (D)-, B.1.617.2 (E)-, and B.1.1.7 (F)-specific IgG titer. At weeks 5 and 7, no significant difference in VoC-specific IgG or IgA titer is observed between TVC-VII and TVC-VIII. Student's *t*-test was used for statistical analysis (ns > 0.05; **P* < 0.05; ***P* < 0.01).

Fig.S15. Comparison of NAb between TVC-VII and TVC-VIII. The serum NAb data in Fig.8F and L were re-analyzed by comparison of WA1 (D614G)-, B.1.617.2-, B.1.1.7-, BA.1-, and BA.4/5-specific NAb titer at doses of 1.2×10^6 PFU (A) and 1.2×10^6 PFU (B). No significant difference VoC-specific NAb between TVC-VII and TVC-VIII is observed. Student's *t*-test was used for statistical analysis ($ns > 0.05$).

In a separate study, we recently showed that another trivalent vaccine (rMeV-BA.1, rMuV-JL1- B.1.617.2, and rMuV-JL2-WA 1) induced broad NAbs against Omicron BA.1, BA.4/5, B.1.617.2, and D641G whereas monovalent vaccine (rMeV-BA.1) only neutralized BA.1 but not BA.4/5 or B.1.617.2 variant in both IFNAR^{-/-} mice (Fig.3H) and hamsters (Fig.7I) (Xu et al., PNAS, 2023). Because the antigenicity of Omicron BA.1 is distinct from the original WA1 and B.1.617.2, it is necessary to include the spike of BA.1 to induce broad neutralizing against D641G, B.1.617.2, and BA.1. Thus, the necessity of including 2 or 3 spikes of variants depends on the antigenic distance of these variants used for trivalent vaccine design. We have discussed this point (please see Discussion section lines 802-818)

Supporting figure: Antigenic distance among variants

Supporting figure: Fig.3H from our recent paper (Xu et al., PNAS, 2023): IFNAR1^{-/-} mice (n = 5) were immunized with a dose (1.5×10^6 PFU) of monovalent (rMeV-BA.1) or trivalent vaccine (rMeV-BA.1-preS-6P, rMuV-JL1-Delta-preS-6P, and rMuV-JL2-WA 1-preS- 6P) via a combination of i.n. and s.c. routes and were boosted three weeks later. Sera at week 7 were used for pseudotype neutralization assay against SARS- CoV- 2 D614G, Delta, Omicron BA.1, or BA.4/5 spike. Data are expressed as the mean of five mice \pm SD. Dotted line indicates the limit of detection (LOD).

Supporting figure: Fig.7I from our recent paper (Xu et al., PNAS, 2023): Golden Syrian hamsters were immunized with a dose (1.5×10^6 PFU) of monovalent (rMeV-BA.1) or trivalent vaccine (rMeV-BA.1-preS-6P, rMuV-JL1-Delta-preS-6P, and rMuV-JL2-WA 1-preS- 6P) via a combination of i.n. and s.c. routes and were boosted three weeks later. Sera at week 7 were used for the pseudotype neutralization assay against SARS- CoV- 2 D614G, Delta, Omicron BA.1, or BA.4/5 spike. Data are the mean of fifteen hamsters \pm SD. Dotted line indicates the limit of detection (LOD).

In summary, the necessity of including spikes of variants in the trivalent vaccine design will depend on the antigenic distance among the variants. We have included new data (new Fig.8, Fig.S12, 13, 14, and 15), Results section lines 552-566, and discussed in the Discussion section (lines 802-818).

3. A critical step for any new modality, is to test against existing approved approaches. It wasn't possible to obtain mRNA vaccines before, but it can be done now. Comparing to mRNA vaccines for immune responses might be critical to advance this platform. If it isn't possible to test against mRNA because of the mouse model, so state in the discussion.

Response: We thank the reviewer for this suggestion. We made many attempts to obtain Moderna and Pfizer mRNA vaccines for research purposes but we cannot get their mRNA vaccines. However, in the literature, many published papers have shown that intramuscularly delivered mRNA vaccine induced strong serum IgG and NAb but was not able to trigger mucosal immune responses (mucosal IgA and lung-resident memory T cell) in animal models and humans (Tang et al., Science Immunology, 2022; Mao et al., Science, 2022; Nickel et al., PloS One, 2022). That is one of the reasons that WHO and the White House called for the development of a next-generation intranasal COVID-19 vaccine. Like several reported intranasal vaccine candidates, our trivalent vaccine candidates are capable of inducing strong serum IgG and NAb as well as mucosal IgA and lung-resident memory T cell immune responses. We have discussed these points (please see Discussion section lines 706-710).

Also, BNT162b2 or mRNA-1273 may not be a good control as these mRNA vaccines expressing preS-2P. However, all of our vaccine candidates used in this manuscript expressing preS-6P. Previously, we showed that preS-6P was more immunogenic than preS-2P (Lu et al., PNAS, 2022).

More detailed comments:

Results

1. Figures 1-3 are not needed in the main body of the paper. The paper should focus on the immunogenicity and protection from viral challenge. The early figures should be used in a different paper on the making the vaccine candidates or put in the supplemental.

Response: Yes, we have moved Figures 1-3 to supplementary figures 1-3. Please see Fig.S1-3.

2. While the use of the trivalent seems to have improved the results of using the monovalent vaccine, what would have been better would be to compare to null 2nd or 3rd viruses. Is there an adjuvant effect?

Response: Please see our above response (Point 1b). There is no adjuvant effect (Fig.S11).

3. Figure 5. What the signal at day 0? Is it always 2 log₁₀? The mucosal responses seem to look the same as the IgA Serum responses in figure 4F. Is this due to blood contamination in the preps?

Response: For Fig.5, we have provided the data at day 0. We used an initial dilution of 1:100 for serum IgG ELISA. Therefore, the detection limit for serum IgG is 2 log₁₀. Our previous studies showed that serum IgG is lower than serum IgA. Thus, we used an initial dilution of 1:2 for serum IgA ELISA thereby

the detection limit for serum IgA is log2. The data points on IgA titer in Fig.5 are different from Fig.4F. They are not from the contamination of samples.

4. Figure 6. The IFN γ responses seem low relative to TNF α . If this is the real response, then perhaps a comparison between other COVID-19 mouse studies would be appropriate in the discussion.

Response: In Fig.6 (now Fig.3), it is true that IFN γ is relatively lower than TNF α in our experiment. We have discussed our results with the papers published by other researchers using MeV or MuV as the vector. In a measles virus-based SARS-CoV-2 vaccine study, IFN γ is relatively lower than TNF α (Hörner et al., PNAS 2020). In another measles virus-based SARS-CoV-2 vaccine study, the percent of CD4+ and CD8+ IFN γ + cells is similar to those of TNF α + cells (Frantz et al., Nature Communications, 2021). **We have discussed this (Discussion section lines 747-758).**

Discussion.

6. There needed to be a discussion of the animal models and vaccine doses, and relevance to humans. Mice are substantially smaller animals and using full human doses is already substantial overkill. In fact, the authors used much higher doses in mice than humans. The authors were using 1e6 PFU, and humans get 2-3 logs lower. Further, the use of IFN- α KO mice to promote replication seems likely to cause interpretation problems when moving to humans. How relevant is the mouse model to humans?

Response: We have discussed the animal model and vaccine doses. We have now shown that 3×10^4 PFU of trivalent vaccine is sufficient to induce a strong immune response in hamsters. For testing the efficacy of live attenuated measles virus and mumps virus vaccine in small animal models (such as mice and hamsters), 10^5 - 10^7 PFU or TCID $_{50}$ of the vaccine virus was used for immunization. The dose used in our study is higher than those used in human vaccine. The natural host of measles virus and mumps virus is human; they replicate robustly in humans. However, the homology of viral entry receptors between mice and humans is low. Therefore, neither measles virus nor mumps virus infects and replicates efficiently in immunocompetent mice. However, A129 mice which lack type I interferon receptor (IFNAR) are susceptible to infection of both measles virus and mumps virus. To date, this is the ONLY mouse model reported to be susceptible to MeV and MuV infection. We agree that A129 mouse has its limitations because it lacks the IFNAR receptor. However, we also used another rodent model (hamsters) for testing vaccine efficacy. Also, the results of the A129 mouse were verified in immunocompetent hamster models. **Please see Discussion section lines 773-788.**

7. There was no discussion about how the hamster results compared to other hamster studies with mucosal vaccines. Are these results similar, better? Bricker, et al, Cell Reports; Braun, et al, Frontiers in Immunology, are some examples.

Response: We have discussed our intranasal vaccine with other studies with mucosal vaccines. **Please see the Discussion section lines 762-771.**

8. Omicron is weaker virus in hamsters than the other challenge viruses. A brief mention of this seems important in interpreting any results on the virus. The authors were focused on the fact that their vaccine protected in the absence of nAbs, but this might have been simply because of the weaker challenge.

Response: Omicron BA.1 has reduced replication and pathogenesis in hamsters because of hACE2 usage. Indeed, Thus, Omicron BA.1 replicates higher titer and is more pathogenic in hACE2 transgenic hamsters (Halfmann et al., Nature, 2022). In our study, we first transduced hamsters with adenovirus expressing hACE2 receptor before the challenge with Omicron BA.1. As shown in Fig.7 and 10, the Omicron BA.1 replicated high titers in lung and nasal turbinate and caused moderate to severe lung pathology in the unvaccinated hamsters. In contrast, the TVC-V or TVC-VI immunized group had 200- and 50- times reductions in viral titer in the lung and nasal turbinate, respectively, and had significantly reduced lung pathology. TVC-V or TVC-VI had a weak NAb against Omicron BA.1. These results suggest that T cell immunity (particularly mucosal resident T cells) may play a role in protection. This is also supported by other researchers (Keeton et al., Nature, 2022; Chandrashekar et al., Cell, 2022; Gao et al., Nature Medicine, 2022). **We have discussed this (see lines 683-701).**

9. Lots of discussion of the benefits of the replicating measles platform, but there was no mention of the fact that Merck dropped their SARS-COV-2 vaccine approach because it didn't work. That might be important for a balanced discussion.

Response: In our recent publications (Zhang et al., Journal of Medical Virology, 2023), we have discussed the possible reasons why Merck failed the clinical trial of their measles virus-based SARS-CoV-2 vaccine candidate. In brief, the poor immunogenicity of Merck's vaccine may be due to two factors: the suboptimal design of their full-length preS-2P antigen and the nonoptimal position (H-L rather than P-M gene junction) chosen to insert the S gene into the MeV genome.

10. There really needs to be a paragraph on the limitations of the results. The studies were complicated and doses were relatively high for the size of the animal.

Response: We have discussed the limitations of the current study. **Please see Discussion section lines 848-868.**

Methods

1. The method described for obtaining the IgA in the lungs is likely to create contamination from the blood, so it cannot be certain that the response is mucosal or simply monomeric IgA from the blood. Typically, the approach is to flush the lungs carefully at harvest with PBS or to flush the circulatory system before harvesting lungs and grinding them up.

Response: We realized there is an error to describe collection of the lung sample for IgA. We actually used BAL for the determination of IgA titer in the lung. This method was also reported in our previous publication (Xu et al., PNAS, 2023).

From the package insert of the MMR vaccine. After reconstitution, each approximately 0.5 mL dose contains not less than 3.0 log₁₀ TCID₅₀ (tissue culture infectious doses) of measles virus; 4.1 log₁₀ TCID₅₀ of mumps virus; and 3.0 log₁₀ TCID₅₀ of rubella virus.

Response: We have discussed the dose. Please see our new results (**Fig.8**) and discussion lines 773-778.

Reviewer #2

Major comments:

1. In figure 3 and 7, a control group immunized with MeV-WA1 + JL1-WA1 + JL2-WA1 is required to support the possible benefit of a TVC vaccine that comprises two or three different S6Ps over a mono-specific vaccine in this model.

Response: Please see our response to Reviewer#1, Point 2. The necessity of comprising two or three different S6Ps will depend on the antigenic distance between the variants used in the trivalent vaccine design. If the antigenic distance is large, including 2 or 3 spikes of variants broadens NAb against variants. If the antigenic distance is small, it may not be necessary for a combination of several spikes. **We have included new data (Fig.8, Fig.S12, 13, 14, 15) and discussed it in the Discussion section line 801-817.**

2. In Fig. 9, a control group immunized with MeV-WA1 + JL1-WA1 + JL2-WA1 is required to support the possible benefit of the TVC-V and -VI over the monospecific MMM-vectored S2P vaccine.

Response: Please see our response to Reviewer#1, Point 2.

3. Most individuals have been vaccinated with MMR in childhood and thus likely have pre-existing immunity against the MMM-vectored Spike vaccine candidates presented in this work. Spike-specific immune responses induced by i.n. immunization with a TVC combination after priming of mice or hamsters by s.c. immunization with MMM should be compared with such responses in non-primed animals.

Response: Several rMeV-based vaccine candidates (e.g. Chikungunya virus, HIV, Zika virus, and Lassa virus) have been successful in phase I or phase II clinical trials. In phase II clinical trial, Reisinger et al. (2018) specifically examined the impact of pre-existing MeV immunity on the efficacy of rMeV-based Chikungunya virus. They found that rMeV-based Chikungunya virus has excellent immunogenicity in the presence of pre-existing immunity against the vector in healthy adults. For MuV vector, we previously showed that the pre-existing MuV immunity has minimal impact on SARS-CoV-2 S-specific antibody induced by rMuV-pres-6P (Zhang et al., PNAS, 2022).

According to the reviewer's suggestion, we have conducted a new animal experiment to examine the pre-existing MMM vector immunity on the efficacy of TVC vaccines (**new Fig.9, Results section lines 567-588; Discussion section lines 676-680**). Briefly, two groups of hamsters (n=5) were immunized subcutaneously (s.c.) with MMM vector (group 1, with pre-existing immunity) or DMEM (group 2, no pre-existing immunity) at week 0. At weeks 3 and 5, these two groups of hamsters were immunized i.n. with TVC (rMeV-WA1 + rMuV-JL1-B.1.617.2 + rMuV-JL2-B.1.1.7) by i.n. route. Serums were collected at weeks 2, 5, 7, 9, and 11 to examine the serum IgG antibody response against WA1, B.1.617.2 and B.1.1.7 by ELISA. At week 2, MuV and MeV-specific NABs were observed in the MMM vector (Group 1) but not DMEM (Group 2) at week 2, and MuV and MeV-specific NABs in Group 1 were higher than those in Group 2 in the following weeks, demonstrating that preexisting NAB against MuV and MeV has been induced. Both groups were immunized with two doses of TVC at weeks 3 and 5. At week 5, Group 1 had approximately 8 times lower WA1, variant B.1.617.2 and B.1.1.7-specific serum IgG than Group 2. At week 7, Group 1 had approximately 1.5 times lower WA1, variant B.1.617.2- and B.1.1.7-specific serum IgG than Group 2.

However, at weeks 9 and 11, there was no significant difference in SARS-CoV-2-specific IgG titer between Groups 1 and 2 ($P>0.05$) (new Fig.9). These results demonstrate that SARS-CoV-2 specific antibody response is delayed in the presence of pre-existing MMM vector immunity but reaches a similar titer at weeks 7 and 9. Therefore, the preexisting MMM immunity has a minimal impact on the SARS-CoV-2-specific antibody.

Fig.9. The preexisting MMM vector immunity does not significantly interfere with SARS-CoV-2-specific antibodies induced by a trivalent vaccine. (A) Immunization schedule. Two groups of female hamsters ($n=5$) were inoculated s.c with 1.2×10^6 PFU of MMM vectors (4×10^5 PFU of rMeV, rMuV-JL1, and rMuV-JL2) or DMEM. Three weeks later, both groups were immunized with 1.2×10^6 PFU of TVC-II. At week 5, both groups were boosted with 1.2×10^6 PFU of TVC-II. (B) MuV-specific NAb response measured by plaque reduction neutralization assay. (C) MeV-specific NAb response measured by plaque reduction neutralization assay. (D-F) WA1 (D), B.1.617.2 (E), and B.1.1.7 (F) -specific serum IgG titers were measured by ELISA. Student's *t*-test was used for statistical analysis (ns > 0.05; ** $P < 0.01$; **** $P < 0.0001$).

4. Line 1088: "... or the same volume of DMEM". Please clarify where the results of this DMEM control immunized group are presented in the Results section.

Response: We have updated all figures to include the normal control group.

Minor comments;

1. Line 191: S6P ... secreted Please clarify whether the S6P transgenes were designed for secretion (lacked the C-terminal transmembrane domain?).

Response: Yes, the preS-6P was designed for secretion. The transmembrane domain and intracellular domain of preS-6P were replaced with foldon.

2. What does “normal” mean in Fig.4 G and Fig.10 A, E, and I?

Response: Normal refers to animals inoculated with DMEM as a normal control group.

3. Line 384: “IFN-g-producing” repeated.

Response: It has been deleted.

4. Line 626: multivalent vaccines to prevent influenza are mentioned as a successful approach. It is, however, generally accepted that this success is limited with vaccine effectiveness of 20-60% and as low as 5 to 39% against the H3N2 component in influenza vaccines (McLean and Belongia 2021). Please provide another example of multivalent vaccines that are relatively successful. Perhaps HPV vaccines?

Response: We have deleted the influenza virus example from the discussion. We have used HPV as an example. Multivalent vaccines have been successful in preventing various infectious diseases, such as human papillomavirus (HPV) vaccines. HPV vaccines are designed to protect against several HPV strains, which include a group of related viruses that can infect the genital area, as well as the mouth and throat (Cheng et al., 2020). The two main HPV vaccines currently in use are Gardasil 9 and Cervarix. Gardasil 9 protects against nine HPV types, including those that are associated with cervical, vulvar, vaginal, and anal cancers, as well as genital warts. The effectiveness of HPV vaccines is notably high, and they have been shown to be up to 90% effective in preventing HPV-related cancers and diseases (Zhai et al., 2016; Harper et al., 2017). Please see the discussion lines 790-796.

5. Line 823: hamsters in each group in Fig.9A instead?

Response: It was corrected.

Reviewer #3

1. Recent reports suggests that as a result of this immune imprinting, the currently utilized bivalent mRNA vaccines have dampened effects against new variants and it has been suggested that instead, monovalent vaccines without WA1 designed against the most recent variants would be better. Thus, the experiment that the authors should conduct to demonstrate that their trivalent strategy is effective and does not suffer from these imprinting problems that the bivalent mRNA vaccines face is to provide a trivalent vaccination (with WA1 and variants) as a booster to animals that have already received two or three doses of a monovalent WA1 vaccine and compare this to providing a monovalent Omicron booster in terms of breadth and potency of response.

Response: As suggested by the reviewer, we have performed a new hamster experiment to determine whether WA1 preS-6P-induced immune imprinting affects the efficacy of the trivalent vaccine. We chose to compare the antibody responses between monovalent rMeV-BA.1 and trivalent TVC-X containing rMeV-BA.1, rMuV-JL1-B.1.617.2, and rMuV-JL2-WA1. Briefly, hamsters in Groups 1-3 were first received two

doses of rMuV-JL2-WA1 at weeks 0 and 2 to induce a strong WA1-specific antibody response (**Fig.10A**), followed by immunization with the third dose of trivalent TVC-IX (Group 1), monovalent rMeV-BA1 (Group 2), and monovalent rMuV-JL2-WA1 (Group 3) vaccine at week 5, respectively. At week 7, hamsters receiving the third booster of monovalent rMeV-BA.1 and monovalent rMuV-JL2-WA1 exhibited significantly lower WA1-specific IgG against B.1.617.2- and BA.1-specific IgG whereas hamsters receiving the third booster of trivalent vaccine had a similar level of serum IgG against WA1, B.1.617.2, and BA.1 spikes (**Fig.10B**). At week 9, serum IgG antibodies reached a similar level against WA1, B.1.617.2, and BA.1 spikes in Groups 1, 2, and 3 (**Fig.10C**) ($P>0.05$). Subsequently, week 9 sera were used in a neutralization assay (**Fig.10D** and **Fig.S16**). In Group 1, WA1 (D614G)-specific NAb titer is the highest, B.1.617.2-specific NAb is the second, and BA.1-specific NAb titer is the lowest, although there was no significant difference among them ($P>0.05$). A similar trend was observed in Groups 2 and 3. In Group 2, WA1 (D614G)-specific NAb titer was significantly higher than B.1.617.2- and BA.1-specific NAb titers ($P<0.0001$). In addition, B.1.617.2-specific NAb titer was significantly higher than BA.1-specific NAb ($P<0.0001$). In Group 3, WA1 (D614G)-specific NAb titer was similar to B.1.617.2-specific antibody ($P>0.05$) but was significantly higher than BA.1-specific NAb ($P>0.05$). In addition, Group 2 (monovalent rMeV-BA.1) induces 4.0, 3.2, and 2.5-fold higher D614G-, B.1.617.2, and BA.1-specific NAb than Group 1 (trivalent vaccine) ($P<0.05$), respectively (**Fig.S16C**). Thus, monovalent BA.1 vaccine is more effective in inducing BA.1-specific NAb than the trivalent vaccine. These results indicate that immune imprinting induced by previous WA1 preS-6P immunization reduces the BA.1-specific antibody response of the subsequent trivalent vaccine immunization. However, it should be noted that the dose of rMeV-BA.1 in the trivalent vaccine is 3 times less than in the monovalent rMeV-BA.1 vaccine, which may also contribute to the lower BA.1-specific antibody induced by our trivalent vaccine.

Finally, we determined whether Groups 1-3 were protected from Omicron BA.1 challenge. None of the challenged groups had significant weight loss compared to the normal control (**Fig.10E**). At day 3 post-challenge, BA.1 viral titer in the lungs of Groups 1-3 was below or near the detection limit whereas challenge control had 5 log₁₀ PFU/g tissue of viral titer in lungs (**Fig.10F**). In addition, BA.1 viral titer in nasal turbinates of Groups 1-2 was near the detection limit whereas Group 3 had approximately 4 log₁₀ PFU/g tissue (**Fig.10G**), although there was no significant difference in the viral titers in the nasal turbinate in Groups 1-3 ($P>0.05$). All three groups were provided with near complete protection against an Omicron BA.1 challenge. Therefore, although immune imprinting induced by previous WA1 preS-6P immunization may compromise the BA.1-specific antibody response of the trivalent vaccine, the trivalent vaccine can provide near complete protection against the BA.1 subvariant.

We have now included these new data (new Fig.10, Fig.S16, Results section lines 589-629) and discussed it (lines 819-834).

Fig.10. The impact of immune imprinting on the efficacy of trivalent vaccines. 20 female hamsters were divided into 4 groups ($n=5$). Groups 1, 2, and 3 received two doses of rMuV-JL2-WA1 (1.2×10^6 PFU) at weeks 0 and 3, and Group 4 received two doses of rMuV-JL2 control. At week 5, Groups 1, 2, and 3 received trivalent, monovalent rMeV-BA.1, and monovalent rMuV-JL2-WA1. (A) WA1-specific IgG at weeks 2 and 5. (B) WA1-, B.1.617.2-, and BA.1-specific IgG at week 7. (C) WA1-, B.1.617.2-, and BA.1-specific IgG at week 9. (D) WA1(D614G)-, B.1.617.2-, and BA.1-specific NAb at week 9. NAb was detected by pseudotype neutralization assay. (E) Body weight changes in hamsters after Omicron BA.1 challenge. Percent of weight at the challenge day was shown. Data are average of 5 hamsters \pm SD. (F) Omicron BA.1 titer in lung of hamsters after BA.1 challenge. (G) Omicron BA.1 titer in nasal turbinate of hamsters after BA.1 challenge. Data were analyzed using two-way ANOVA and one-way ANOVA (ns > 0.05; ** $P < 0.01$; *** $P < 0.001$; **** $P < 0.0001$).

Group 1: rMuV-JL2-WA1 (1st immunization) + rMuV-JL2-WA1 (2nd immunization) + TVC-VIX (rMeV-BA.1 + rMuV-JL1-B.1.617.2 + rMuV-JL2-WA1) (3rd immunization)
 Group 2: rMuV-JL2-WA1 (1st immunization) + rMuV-JL2-WA1 (2nd immunization) + monovalent rMeV-BA.1 (3rd immunization)
 Group 3: rMuV-JL2-WA1 (1st immunization) + rMuV-JL2-WA1 (2nd immunization) + rMuV-JL2-WA1 (3rd immunization)

Fig.S16. The impact of immune imprinting on induction of D614G-, B.1.617.2-, and BA.1-specific NAb of a trivalent vaccine. The NAb data in Fig.10D were re-analyzed by comparison of WA1(D614G) (A), B.1.617.2 (B), and BA.1 (C) specific NAb. Data were compared using Student's *t*-test (ns > 0.05; ***P* < 0.05; ****P* < 0.01; *****P* < 0.001; ******P* < 0.0001).

Major comments:

- It would be helpful to the scientific community to provide the sequences of the vaccines as a supplement or deposited to GenBank.

Response: We have provided all spike sequences in the supplementary figure (Table S3).

- It is noted that there are 48 possible trivalent vaccine combinations in lines 254-254, but there is no explanation provided for why the six chosen combinations were chosen. To this reviewer it is not very clear why these specific combinations were chosen. Perhaps the authors can add their rationale for each of the combinations.

Response: We now listed all the new recombinant viruses we made in this manuscript (Table S1). It is not possible to test all 48 trivalent vaccine combinations. The rationales for choosing the six combinations are summarized in Table S2. The goal is to choose a trivalent MMM vaccine expressing 2 or 3 different preS-6P proteins.

Table 1: Recombinant viruses generated in this study

Recombinant virus	Targeted antigen	Origin
rMeV-WA1	Expresses preS-6P gene of SARS-CoV-2 WA1	Zhang et al., JMV, 2023
rMeV-B.1.351	Expresses preS-6P gene of B.1.351 VoC	This study
rMeV-B.1.1.7	Expresses preS-6P gene of B.1.1.7 VoC	This study
rMeV-B.1.617.2.	Expresses preS-6P gene of B.1.617.2.VoC	This study
rMeV-BA.1	Expresses preS-6P gene of Omicron BA.1	Xu et al., PNAS, 2023
rMuV-JL2-WA1	Expresses preS-6P gene of SARS-CoV-2 WA1	Zhang et al., PNAS, 2022
rMuV-JL2-B.1.351	Expresses preS-6P gene of B.1.351 VoC	This study
rMuV-JL2-B.1.1.7	Expresses preS-6P gene of B.1.1.7 VoC	This study
rMuV-JL2-B.1.617.2	Expresses preS-6P gene of B.1.617.2.VoC	This study
rMuV-JL1-WA1	Expresses preS-6P gene of SARS-CoV-2 WA1	This study
rMuV-JL1-B.1.1.7	Expresses preS-6P gene of B.1.1.7 VoC	This study
rMuV-JL1-B.1.617.2	Expresses preS-6P gene of B.1.617.2.VoC	Xu et al., PNAS, 2023

Table 2: Trivalent vaccine candidates tested in this study

Vaccine candidate	Composition	Rationale	Experiment
Trivalent Vaccine Candidate I (TVC-I)	rMuV-JL2-WA1, rMuV-JL2-B.1.1.7, and rMeV-B.1.351	3 viruses expressing 3 different preS-6P proteins	Animal Experiment 1
TVC-II	rMuV-JL2-WA1, rMuV-JL2-B.1.1.7, and rMeV-WA1	rMeV and rMuV-JL2 expressing WA1, and rMuV-JL2 expressing B.1.1.7	Animal Experiment 1
TVC-III	rMuV-JL1-WA1, rMuV-JL2-B.1.617.2, and rMeV-B.1.351	3 viruses expressing 3 different preS-6P proteins	Animal Experiment 2
TVC-IV	rMeV-B.1.617.2, rMuV-JL1-B.1.1.7, and rMuV-JL2-WA1	3 viruses expressing 3 different preS-6P proteins	Animal Experiment 3
TVC-V	rMuV-JL1-WA1, rMuV-JL2-B.1.1.7, and rMeV-WA1	rMeV and rMuV-JL1 expressing WA1, and rMuV-JL2 expressing B.1.1.7	Animal Experiment 4
TVC-VI	rMuV-JL1-WA1, rMuV-JL2-B.1.617.2, and rMeV-WA1	rMeV and rMuV-JL1 expressing WA1, and rMuV-JL2 expressing B.1.617.2	Animal Experiment 4
MMM vector	rMeV, rMuV-JL1, and rMuV-JL2.	3 vectors	Animal Experiments 1-4
Monovalent	rMuV-JL2-WA1	Monovalent	Animal Experiments 1, 3, and 5
Monovalent	rMuV-JL2-B.1.617.2	Monovalent	Animal Experiment 4
TVC-VII	rMuV JL2-WA1 + rMuV-JL1-WA1 + rMeV-WA1	Comparison of 3 viruses expressing WA1 and 3 viruses expressing 3 different spikes	Animal Experiment 6
TVC-VIII	rMeV-WA1 + rMuV- JL1- B.1.617.2+ rMuV-JL2-B.1.1.7	Comparison of 3 viruses expressing WA1 and 3 viruses expressing 3 different spikes	Animal Experiments 6 and 7
TVC-IX	rMeV-BA.1 + rMuV-JL1-B1.617.2 + rMuV-JL2-WA1	Immune imprinting experiment	Animal Experiment 8
Monovalent	rMeV-BA.1	Immune imprinting experiment	Animal Experiment 8

- There is a focus on the current generation of vaccines not providing robust mucosal immunity and consequently the described vaccines were developed, but it would be helpful if such a group (e.g., vaccinated with BNT162b2 or mRNA-1273) was included as a comparator to know how much greater the mucosal immunity generated by these vaccines is.

Response: Please see our response to Reviewer #1. We were not able to obtain BNT162b2 or mRNA-1273 vaccines. However, many papers have shown that mRNA vaccine cannot trigger mucosal immunity (Tang et al., *Science Immunology*, 2022; Azzi et al., *eBioMedicine* 2022; Bladh et al., *Lancet Microbe*, 2023). For example, Tang et al., compared the respiratory antibody response and tissue-resident memory T and B cell responses in individual who had recovered from COVID-19 and those who had received two doses mRNA vaccinated individuals (BNT162b2 or mRNA-1273). The results showed that, in contrast to natural infection, mRNA vaccination did not provoke robust IgA responses and induce significant SARS-CoV-2-specific B and T cell memory in the respiratory tract. We have discussed this point (Discussion section lines 706-710).

Also, BNT162b2 or mRNA-1273 may not be a good control as these mRNA vaccine expressing preS-2P. However, all of our vaccine candidates expressing preS-6P. Previously, we showed that preS-6P was more immunogenic than preS-2P (Lu et al., *PNAS*, 2022).

- Figures 2C and 3C: Is the antibody that is being used in these blots specific for S or does it also bind S1 and S2? Are the other bands shown here S1/S2? If so, why do the three vectors have differing cleavage ratios?

Response: The antibody used in Western blot specifically targets the RBD domain of S protein. All preS-6P used in this study is a soluble, stabilized prefusion form with a mutated furin cleavage site. Although the majority of protein detected is full-length uncleaved preS-6P (approximately 180 KDa), a small amount of preS-6P may be cleaved. For MuV-JL1 constructs (Fig.3C), we observed more cleaved products at 96h. We believe that this is because MuV-JL1 constructs had much more secreted preS-6P at 96 h (the bands are much stronger compared to 72h).

- In the experiments shown in Figure 4, 7, and 9, it seems unexpected that the monovalent vaccine, which has 1.5 or 3 times the amount of this antigen than the trivalent vaccines, still has a lower autologous response. E.g., in Figure 4, why does rMuV-JL2-WA1 alone not have a higher WA1-directed response than the two trivalent groups? Perhaps the authors can add an explanation for this to the Discussion.

Response: We have discussed this. The trivalent group consists of three viruses: MeV, MuV-JL1, and MuV-JL2 whereas the monovalent group only includes MuV-JL2. Although the primary target of both MeV and MuV is the respiratory tract, they have different host tropisms and may infect different tissues, organs, and cell types when they are delivered intranasally. It is possible that MeV, MuV-JL1, and MuV-JL2 carrying

three preS-6P had synergetic effects thereby inducing stronger antibody and T cell immune responses than monovalent rMuV-JL2-WA1 alone. We have discussed this lines 656-660.

- Figures 4, 5, 7, 9: It is confusing and makes it difficult to compare to have IgG in log10 but IgA in log2 units, please place them on the same scales.

Response: Because serum IgG level is much higher than serum IgA, the initial dilution of serum for IgG is 1:100 whereas the initial dilution of serum for IgA is 1:2. That is the reason why we used log10 and log2 as the unit. As recommended by the reviewer, we have used log10 as a scale for all IgG and IgA titer.

- Figure 7: A vector control group is mentioned in the text but not shown in any of the panels. Why is there seemingly no difference between the low and high dose of vaccine?

Response: We have added the data for the vector group for the figures. The vector control group did not induce IgG and IgA responses against spike proteins. In this experiment, we did not observe any difference in antibody responses between the low dose (3×10^5 PFU) and the high dose (1.2×10^6 PFU). The low dose (3×10^5 PFU) of trivalent vaccine is sufficient to induce high levels of IgG and IgA responses against spike protein. Similarly, our new hamster immunization experiment showed that a 10-fold lower dose (3×10^4 PFU) induced similar levels of serum IgG compared to doses of (3×10^5 and 1.2×10^6 PFU).

- Figure 8: Why does the monovalent group have less CD4+ Trms than the trivalent group? In particular, why would it have less than the low dose trivalent group?

Response: The trivalent group consists of three viruses: MeV, MuV-JL1, and MuV-JL2 whereas the monovalent group only includes NuV-JL2. Although the primary target of both MeV and MuV is the respiratory tract, they have different host tropisms and may infect different tissues, organs, and cell types when they are delivered intranasally. Thus, the trivalent vaccine may induce a higher CD4+ Trms than the monovalent vaccine. In this experiment, a low dose (3×10^5 PFU) of trivalent vaccine included similar levels of T cell immune responses compared to the high dose (1.2×10^6 PFU) of trivalent vaccine. A dose of 3×10^5 PFU is sufficient to induce a higher level of Trms. We have discussed this (lines 656-660 and lines 773-784).

Minor comments:

- Figures 1D, 2D, 3D, S1, S2, S3: Please add a scale bar.

Response: A scale bar has been added.

- Figure 4G: What is the blue “ns”?

Response: “ns” stands for “not significant”

- Figure 4: This reviewer suggests to maintain consistency in coloring throughout the panels; e.g., the green in A through F is different than the green in G and H.

Response: The color issue has been fixed.

- Figure 5D, 9H: It may be apt to label Alpha, Beta, and Delta as B.1.1.7, B.1.351, and B.1.617.2 given that they are labeled as such in all other panels.

Response: These have been fixed.

- Figure 5E, 5F, 5G: The y-axes should be kept the same to allow for easier comparisons between panels.

Response: It has been fixed.

- Figure 10K: The y-axis should be kept the same as other panels to allow for easier comparisons between panels.

Response: It has been corrected.

- Figure 11: This figure could perhaps be moved to the supplement.

Response: The old figure 11 has been moved to a supplementary figure (new Fig.S10).

- Lines 893-900: Please provide the accession numbers for the sequences of the viruses used.

Response: Accession numbers have been provided.

- Lines 992-1012: The mutations that were introduced to produce the variant pseudoviruses and spikes should be noted (as not everyone has used the same mutations for the variants, particularly Omicron).

Response: These information has been provided. The mutations in spike proteins of B.1.617.2, B.1.1.7, B.1.351, Omicron BA.1, and Omicron subvariant BA.4/5 were described in co-author Shan-Lu Liu publications (see lines 1178-1179). References were cited.

- Line 1178: A reference seems to be missing here.

Response: A reference has been added.

- References 38 and 60 appear to be the same.

Response: It has been corrected.

REVIEWERS' COMMENTS

Reviewer #1 (Remarks to the Author):

The authors addressed my concerns about the paper and have done substantial additional work. I have no further concerns.

Reviewer #2 (Remarks to the Author):

Thank you for addressing the reviewer comments. The results of the newly performed hamster immunization experiments are clarifying the remarks that were made.

One minor point: Line 319: BAL performed instead of (lung) homogenization?

Reviewer #3 (Remarks to the Author):

The authors have conducted multiple additional animal studies in this revision to address the issues that were raised. The authors should be commended for the depth of work that they have conducted in this manuscript.

The additional experiment examining immune imprinting appears to demonstrate quite well that this trivalent strategy faces this same issue as the now-discontinued bivalent mRNA vaccines. While the authors' note that the trivalent vaccine has 1/3 of the antigen compared to monovalent is well-taken and is plausible, it seems unlikely to be the issue given that in their other additional experiment looking at dosing, they demonstrate that there is essentially a plateau effect between the highest dose (used for this experiment) and the medium dose (which has 1/4 of the antigen of the high dose). Some optimization of variants used would be required if this trivalent vaccine strategy were to be used clinically in a population that has already been heavily immunized. But I think that is outside the scope of this work and it is acceptable to note that this is a limitation of this strategy as the authors have done.

A minor note that there is a typo through some of the figures and text in which the delta variant is denoted as "B.1.167.2" but it should be "B.1.617.2".

Response to reviewers' comments

- (1) Reviewer#1: Thank you for the positive comments.
- (2) Response to Reviewer #2: Thank you. In Line 319, "lung homogenization" has been changed to "BAL". This has been stated in the legend of Fig.2.
- (3) Response to Reviewer#3: Thank you for reviewer's positive comments on our additional experiments on doses and immune imprinting. We have discussed the limitation of this study in the Discussion section. We have checked carefully for the typos through the manuscript.